# Unprotected peptide macrocyclization and stapling via a fluorine-thiol displacement reaction

Md Shafiqul Islam[1], Samuel L. Junod[2], Si Zhang[1], Zakey Yusuf Buuh[1], Yifu Guan[1], Mi Zhao[1], Kishan H. Kaneria[1], Parmila Kafley[1], Carson Cohen[1], Robert Maloney[1], Zhigang Lyu[1], Vincent A. Voelz[1], Weidong Yang[2] & Rongsheng E. Wang [1 ✉]

We report the discovery of a facile peptide macrocyclization and stapling strategy based on a fluorine thiol displacement reaction (FTDR), which renders a class of peptide analogues with enhanced stability, affinity, cellular uptake, and inhibition of cancer cells. This approach enabled selective modification of the orthogonal fluoroacetamide side chains in unprotected peptides in the presence of intrinsic cysteines. The identified benzenedimethanethiol linker greatly promoted the alpha helicity of a variety of peptide substrates, as corroborated by molecular dynamics simulations. The cellular uptake of benzenedimethanethiol stapled peptides appeared to be universally enhanced compared to the classic ring-closing metathesis (RCM) stapled peptides. Pilot mechanism studies suggested that the uptake of FTDR-stapled peptides may involve multiple endocytosis pathways in a distinct pattern in comparison to peptides stapled by RCM. Consistent with the improved cell permeability, the FTDR-stapled lead Axin and p53 peptide analogues demonstrated enhanced inhibition of cancer cells over the RCM-stapled analogues and the unstapled peptides.

---

[1] Department of Chemistry, Temple University, 1901 N. 13th Street, Philadelphia, PA 19122, USA. [2] Department of Biology, Temple University, 1900 N. 12th Street, Philadelphia, PA 19122, USA. ✉email: rosswang@temple.edu

Protein–protein interactions (PPIs) regulate important molecular processes including gene replication, transcription activation, translation, and transmembrane signal transduction[1–3]. Aberrant PPIs have been consequently implicated in the development of diseases, such as cancer, infections, and neurodegenerative diseases[3,4]. Targeting PPIs has since emerged as a promising therapeutic strategy that specifically inhibits specific molecular pathways without compromising other functions of the involved proteins[5]. Yet this avenue is challenging due to the generally flat, shallow, and extended nature of PPI interfaces[1,6,7]. In this regard, efficient mimicking of peptides involved in PPIs is a long-standing direction in PPI inhibitor development[6]. Driven by entropy and the interactions with water, short peptides are mostly unstructured random coils in aqueous solutions[8]. Given that over 50% of PPIs involve α-helices[9,10], one common approach uses chemical stapling to restore and stabilize the bioactive helical conformation of short peptides[11]. By introducing side-chain to side-chain crosslinking of residues positioned at the same face (an $i, i+4$ or $i, i+7$ fashion) of the α helix, the structures of peptides are locked in folded conformation due to the reduced entropy penalty[11].

Using olefin-containing unnatural amino acids and ring-closing metathesis (RCM) mediated chemical crosslinking, Grubbs, Verdine, and Walensky et al. have done seminal work to develop RCM-stapled peptides as a promising class of therapeutics[12–18]. The hydrocarbon stapled peptides not only maintained the desired tertiary structures and the associated targeting specificities but also possessed increased binding affinity and protease resistance[1,15,18]. One hydrocarbon stapled peptide, a lead p53 mimic (ALRN-6924), has been in clinical testing to treat a diverse set of tumours including acute myeloid leukaemia (AML)[2]. Yet, RCM-based stapling necessitates the use of metal-based catalysts that were sometimes incompatible with other functional groups present in peptides such as a thiourea moiety[19]. The increased hydrophobicity brought by hydrocarbon linkers may also incur issues in targeting specificity and aqueous solubility[19]. To date, a number of other stapling strategies based on different crosslinking chemistry have been established[7,11,20–27], each bearing different strengths and weaknesses[7,24,25]. For instance, many approaches required additional catalysts such as metal complexes or photoinitiators. Direct crosslinking based on bromo or iodo-alkyl/benzyl chemistry, on the other hand, could produce over-alkylated species due to their increased cross-reactivity[20–22,25]. Most strategies hinge on reactions with native amino acid residues such as cysteines[23,26,28–31], which however often play essential roles in protein–protein intermolecular interactions and could be also necessary to retain the desired structures and functions[32–34]. More importantly, the vast majority of stapled peptides still had limited cell permeability[35]. As a result, the development of stapled peptides with designable cell permeability remains challenging and requires a delicate balance between positive charges, hydrophobicity, alpha-helicity, and staple position[35–37].

Thus, novel strategies capable of stapling unprotected peptides in a straightforward, chemoselective, and clean manner, as well as promoting cellular uptake are highly sought[11]. Unlike other readily reacting α-haloacetamides, a fluoroacetamide functional group has been considered biologically inert due to the poor leaving capability of fluorine[38]. A number of probes consisting of the radiolabeled [18F]fluoroacetamide have been thereby developed as positron emission tomography (PET) agents for in vivo diagnostics[39–41], suggesting the biorthogonality of the fluoroacetamide functional group. Nevertheless, fluoroacetamide was recently incorporated into proteins as the side-chain of an unnatural amino acid, and was revealed to react with the thiol group of cysteine within protein-confined close proximity[38]. Although driven by the proximity effect, this type of reaction may

happen intermolecularly if at higher concentrations. Further, with some more nucleophilic sulfhydryl functional groups, fluorine could be more efficiently displaced. Together with the biorthogonality of fluoroacetamide, this suggests that a fluorine–thiol displacement reaction (FTDR) happening intermolecularly could be highly selective and would satisfy the requirements of a "click" chemistry reaction, making it potentially attractive in the context of bioconjugations.

In this work, we report the discovery of a fluorine displacement reaction using different thiol-containing linkers, which could be used for the mild functionalization of unprotected peptides. The reaction could selectively proceed with benzenedimethanethiol linkers in the presence of intrinsic cysteines. This versatile approach allowed the facile preparation of constrained macrocyclized peptides of different linker sizes, and led to the identification of stapled peptides that possessed improved target binding versus wild type peptides, and much enhanced cellular permeability compared to hydrocarbon stapled peptides. The peptide analogues stapled via FTDR using the benzenedimethanethiol linkers in general exhibited three to nine-fold enhanced cellular uptake than the hydrocarbon stapled ones. The corresponding Axin and p53 peptide analogues also showed enhanced growth inhibition of cancer cell growth as a result of increased cellular uptake, demonstrating the potential of FTDR-stapled peptides as probes for targeting intracellular compartments.

## Results

**FTDR-based coupling with model compound/peptides.** We started first by using the reported fluoroacetamide **1** as a model compound that is UV active and allows the facile monitoring of reaction progress on LC–MS[42]. Evaluation of its reaction with common nucleophiles such as methyl hydrazine and cysteine in a mildly basic Tris (tris(hydroxymethyl)-aminomethane) buffer (Supplementary Fig. 1.1, Supplementary Fig. 1.3) revealed that although there was no reaction between fluoroacetamide and methyl hydrazine after 12 h of incubation at 37 °C, ~40.4% of fluoroacetamide had been converted by highly concentrated cysteine to the fluorine-displaced adduct. This suggests that the fluoroacetamide functional group reacts specifically to thiols at high concentrations. We then attempted the fluorine thiol displacement reaction with benzyl thiol[43,44], and observed a further enhanced reaction (Supplementary Figs. 1.1 and 2), rendering a relative yield of 53%. Given the recently reported fluorine–π interactions[45,46], fluorine of the fluoroacetamide side chain may interact with the aromatic benzyl ring of benzyl thiol, thereby attracting thiol to a close proximity, facilitating the displacement of the fluorine. With this encouraging result, we further attempted the reaction on a protected amino acid building block **9** which has a natural amino acid backbone but possesses a fluoroacetamide functional group in the side chain (Fig. 1a). Consistent with the previously observed FTDR results, the alpha fluoride was efficiently replaced by benzyl thiol, giving the desired product **17** at a yield of 73% after purification. We then moved on to evaluate the chemoselectivity of this FTDR as a general macrocyclization method using a seven amino acid long model peptide **18** that bears multiple unprotected functional groups (Fig. 1b). Given the documented ability of the sulfur to stabilize the negative charge[26], 1,4-benzenedimethanethiol, a previously reported bifunctional benzyl thiol linker[44], was pre-activated by deprotonation of di-thiols with a less than a stoichiometric amount of base (NaOH) and was then incubated with the model peptide **18** in a mixture of water/DMF. The reaction was monitored by LC–MS analysis (Supplementary Fig. 3), which indicated significant conversion of the starting material and the mono-substituted intermediate into the macrocyclized peptide

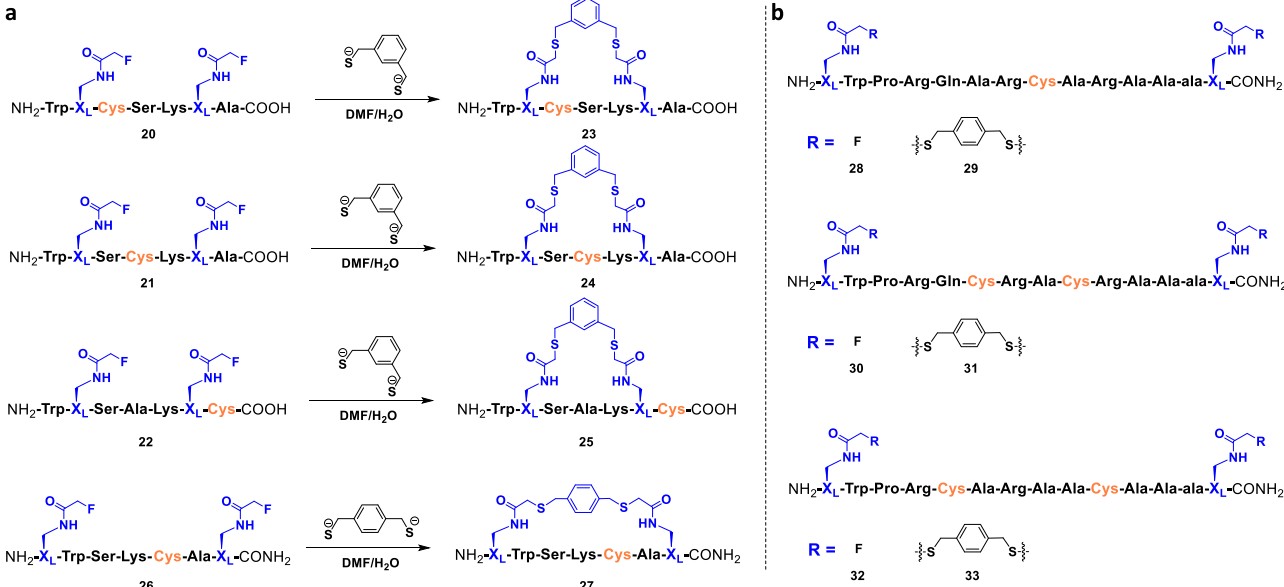

**Fig. 1 Fluorine thiol displacement reactions (FTDRs). a** A model reaction between benzyl thiol and the α-fluoroacetamide containing amino acid building block **9**. **b** A model macrocyclization carried out between unprotected model peptide **18** and a commercially available linker 1,4-benzenedimethanethiol based on FTDR (pH = 9.0–9.5). The α-fluoroacetamide containing amino acid $X_L$ is highlighted in blue.

**Fig. 2 FTDR-based coupling with cysteine-containing model peptides. a** FTDR-based macrocyclization reactions on model peptides **20–22** and **26** which have cysteines and $X_L$ at different positions. **b** FTDR-based stapling of peptides (**28**, **30**, and **32**) which consist of internal cysteines at varied positions and terminal $X_{LS}$. The commercially available linker benzenedimethanethiol was used for these series of model cyclization and stapling reactions, and the final reaction pH was 9.0–9.5. Cysteine is highlighted in orange and $X_L$ is highlighted in blue.

product **19**. The observed transformation was >90% completed after 12 h, and the cyclization yield was ~62% after HPLC purification. Next, we prepared another series of model peptides that possess amino acids with other unprotected side chains such as serine and cysteines (Fig. 2a).

To explore the reaction compatibility with cysteines, peptides **20–22** were incorporated with unprotected cysteines at different sites. FTDR-based coupling with another benzyl thiol linker, 1,3-benzenedimethanethiol, proceeded smoothly for the above cysteine-containing peptides, generating the fluorine-displaced macrocyclization as the major product **23–25** (Supplementary Fig. 4). The transformation was >90% within 12 h reaction time and rendered the desired cyclization products with cyclization yields >70% (Supplementary Table 1). Notably, the cyclization with fluoroacetamide-chained amino acids at N- and C-terminals also proceeded efficiently, as demonstrated by the conversion of peptide **26** to **27** (Fig. 2a, Supplementary Table 1). Given the

observed conversion with benzenedimethanethiol, the FTDR-based coupling appears to be chemoselective and orthogonal to functional groups in natural amino acid side chains such as carboxylic acids, amines, alcohols, phenolic alcohols, and alkanethiols.

**Substrate scope with various linkers**. We then decided to evaluate the FTDR-based cyclization on varied peptides of interesting chemical and biological properties. Structurally, a 14-mer rationally designed peptide has been recently macrocyclized at both N- and C-terminal to stabilize the whole peptide as a full α-helix[47]. We wondered if FTDR-based N-, C-terminal macrocyclization could promote α-helical peptide folding as well. As shown in Fig. 2b, peptides **29**, **31**, and **33** have been successfully capped by FTDR-mediated cyclization even in the presence of one or more cysteines. Evaluation by circular dichroism (CD) (Supplementary Fig. 5) confirmed that after terminal

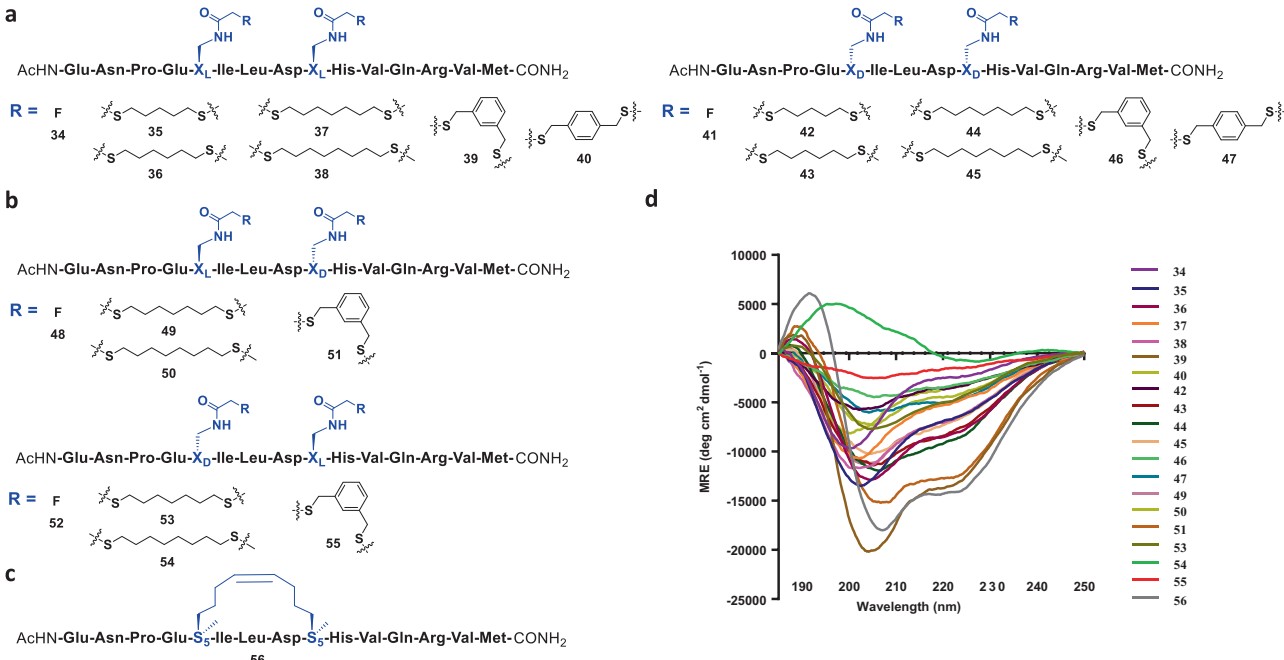

**Fig. 3 FTDR-based coupling between an unprotected Axin peptide analogue and various dithiol linkers with i, i + 4 linkage. a** Coupling with the unprotected analogue **34** that has both L-fluoroacetamide substrates and the analogue **41** that has both D-fluoroacetamide substrates, respectively. **b** Coupling with the unprotected analogue **48** that has L-fluoroacetamide and D-fluoroacetamide (N → C direction), and analogue **52** that has D-fluoroacetamide and L-fluoroacetamide (N → C direction), respectively. **c** The reported Axin peptide analogue **56** was prepared based on ring-closing metathesis. **d** Circular dichroism (CD) spectra of all peptide analogues. Coupling amino acids $X_L/X_D/S_5$ were highlighted in blue.

macrocyclization all three peptides obtained stable α-helical conformations as evidenced by a positive band at 195 nm and two negative bands at 208 and 222 nm[48]. On the contrary, peptides prior to stapling (**28**, **30**, **32**) only displayed the typical signs of random coils with a negative band around 195 nm and a positive band around 215 nm. These results inspired us to further explore unprotected FTDR macrocyclization in peptide stapling in terms of substrate scope and stapling linkers.

The Axin mimetic analogue then caught our attention as it was a classic peptide used for stapling[14,23], and its blocking of the Axin-β-catenin PPI has been shown to inhibit the Wnt signalling pathway that is crucial for the development of colorectal cancers and AML[14,49–51]. A pair of fluoroacetamide-containing amino acids combination of possible chirality (**15** for $X_L$, **16** for $X_D$, see Supplementary Information including Supplementary Fig. 1.2. for detailed synthesis) were thereby incorporated into the i and i + 4 positions by Fmoc-based solid-phase synthesis to render the 15 amino acid long fluoroacetamide containing Axin peptide analogues ($X_L X_L$ for peptide **34**, $X_D X_D$ for **41**, $X_L X_D$ for **48**, and $X_D X_L$ for **52**) (Fig. 3a, b). Using the optimized reaction conditions as mentioned above, we attempted FTDR-based macrocyclization on these peptides with aliphatic dithiols of various lengths or more rigid and reactive benzenedimethanethiol linkers. The reaction mixtures had a final pH of 9.0–9.5, which together with the mild reaction temperature (37 °C) should lead to minimal epimerization[52,53]. As demonstrated in Fig. 3, most linkers can cyclize fluoroacetamide-containing peptides with satisfactory yields (Supplementary Table 1). Efficient macrocyclization was observed on both $X_L$ (**34**) or $X_D$ (**41**) enantiomer combination with aliphatic linkers from 5-carbon to 8-carbon long, and benzenedimethanethiol linkers at meta or para positions. The oppositely angled fluoroacetamide-containing side chains in the $X_L, X_D$ (**48**) or $X_D, X_L$ (**52**) substrates can be only crosslinked by 7-carbon and 8-carbon long aliphatic linkers as well as 1,3-benzenedimethanethiol as the only tolerated aromatic linker. For all the substrates being tested, macrocyclization appeared to

proceed more efficiently than those in small molecule model reactions, presumably due to the complete deprotonation of thiols in advance that ensured the generation of more nucleophilic benzyl thiolate anions[54], and the potentially enhanced fluorine–π interactions with two equivalents of fluoroacetamide in each peptide. Taken together, FTDR-based coupling represents an efficient macrocyclization approach in synthesizing cyclic unprotected peptides with flexible linker choices.

**Linker and chirality requirement in FTDR-based i, i + 4 stapling.** With the library of cyclized Axin peptide analogues on hand, we went on to evaluate the stapling effect of these dithiol linkers crosslinked at the i, i + 4 positions. As a positive control, we prepared the RCM-stapled Axin peptide analogue **56** (Fig. 3c) that had a reported helicity of around 51%[14]. CD experiments were performed to measure the alpha helicity of these peptides (Fig. 3d). As demonstrated in Table 1, the highest helicity was achieved with peptides **39** and **51**, with a mean value of 46% and 44%, respectively, which were close to the measured helicity of the RCM stapled control **56**. Although **39** (L,L) and **51** (L,D) possess different substrate chiralities, both analogues were stapled well by the 1,3-benzenedimethanethiol linker, suggesting this aromatic linker could universally promote the alpha helicity of fluoroacetamide-containing peptides at i, i + 4 linkage. In comparison, cyclized peptides with chirality combinations of D,D (**42–47**) or D,L (**43–55**) all failed to display enhanced alpha helicity to a comparable extent, even for the ones (**46** with a helicity of 13%, **55** with a helicity of 8%) stapled by the 1,3-benzenedimethanethiol linker. Consistent to the literature report[55], substitution with both D-configured amino acids at i, i + 4 positions largely destabilized the intrinsic helix conformation. Yet, the inclusion of D-amino acids at the i position has been well documented, with the resulting D,L crosslinked peptide substrates usually inheriting enhanced stabilization of alpha

**Table 1 Summary of the helicity and the macrocyclization yields of the stapled peptides.**

| Peptide | Fluoro substrates | Helicity (%) | Yield (%) | Peptide | Fluoro substrates | Helicity (%) | Yield (%) |
|---|---|---|---|---|---|---|---|
| **29** | $L_N, C^L$ | 24 | 22 | **57** | $L_{i,\,i+4}L$ | 41 | 64 |
| **31** | $L_N, C^L$ | 21 | 23 | **58** | $L_{i,\,i+4}D$ | 39 | 61 |
| **33** | $L_N, C^L$ | 23 | 27 | **59** | $D_{i,\,i+4}L$ | 13 | 58 |
| **35** | $L_{i,\,i+4}L$ | 24 | 55 | **60** | $L_{i,\,i+4}L$ | 50 | 59 |
| **36** | $L_{i,\,i+4}L$ | 29 | 59 | **61** | $L_{i,\,i+4}D$ | 45 | 51 |
| **37** | $L_{i,\,i+4}L$ | 18 | 49 | **62** | $D_{i,\,i+4}L$ | 9 | 56 |
| **38** | $L_{i,\,i+4}L$ | 25 | 52 | **63**[a] | $S_5, S_5$ | 52 | 51 |
| **39** | $L_{i,\,i+4}L$ | 46 | 63 | **70** | $L_{i,\,i+4}L$ | 84 | 64 |
| **40** | $L_{i,\,i+4}L$ | 17 | 52 | **72** | $L_{i,\,i+4}D$ | 85 | 61 |
| **42** | $D_{i,\,i+4}D$ | 14 | 58 | **73**[a] | $S_5, S_5$ | 88 | 37 |
| **43** | $D_{i,\,i+4}D$ | 28 | 62 | **75** | $LI_{,\,i+4}L$ | 74 | 74 |
| **44** | $D_{i,\,i+4}D$ | 31 | 60 | **77** | $L_{i,\,i+4}D$ | 70 | 73 |
| **45** | $D_{i,\,i+4}D$ | 25 | 55 | **78**[a] | $S_5, S_5$ | 73 | 42 |
| **46** | $D_{i,\,i+4}D$ | 13 | 59 | **80** | $L_{i,\,i+7}L$ | 36 | 65 |
| **47** | $D_{i,\,i+4}D$ | 19 | 65 | **81** | $L_{i,\,i+7}L$ | 42 | 69 |
| **49** | $L_{i,\,i+4}D$ | 25 | 54 | **82** | $L_{i,\,i+7}L$ | 61 | 74 |
| **50** | $L_{i,\,i+4}D$ | 14 | 43 | **83** | $L_{i,\,i+7}L$ | 16 | 55 |
| **51** | $L_{i,\,i+4}D$ | 44 | 58 | **85** | $L_{i,\,i+7}D$ | 40 | 69 |
| **53** | $D_{i,\,i+4}L$ | 19 | 39 | **86** | $L_{i,\,i+7}D$ | 37 | 71 |
| **54** | $D_{i,\,i+4}L$ | 5 | 50 | **87** | $L_{i,\,i+7}D$ | 64 | 79 |
| **55** | $D_{i,\,i+4}L$ | 8 | 51 | **88** | $L_{i,\,i+7}D$ | 18 | 55 |
| **56**[a] | $S_5, S_5$ | 48 | 73 | **89**[a] | $R_8, S_5$ | 65 | 58 |

Bold indicates peptide numbering.
[a]Control peptides stapled by ring-closing metathesis.

helixes and improved binding affinity in comparison to the L,L crosslinked peptides[20,55–58]. On the other hand, there was little observation of D-amino acid's beneficial effects at the $i + 4$ position either, as most crosslinked L,D analogues resulted in a negative effect to the peptides' alpha helix conformation[56,58]. Nevertheless, D-propargylglycine was once substituted at $i + 4$[27], and compared to its L-enantiomer, Copper-mediated Huisgen 1,3-dipolar cycloaddition between L-azido norleucine and D-propargylglycine at $i, i + 4$ positions resulted in much less distortion to the peptide backbone conformation[27]. These suggested that for stapling-induced alpha helical stabilization the chirality requirements of substrate side chains could largely depend on the chemistry used for stapling, and specifically the chemical structures desired for both side chains and crosslinkers.

In order to understand more of the structural information of the FTDR-stapled Axin peptide analogues, we subsequently prepared their corresponding central portions (**57** as a surrogate for **39**, **58** for **51**, and **59** for **55**, Fig. 4a, see SI for detailed characterizations). Consistent with the reported studies that heptapeptides are sufficient in length for stapling and functioning[59,60], peptides **57–58** retained most of the α-helical folding as shown in CD characterization (Fig. 4b, Table 1). A detailed nuclear magnetic resonance (NMR) study was carried out to analyse the conformational behaviours of these peptide analogues (Supplementary Table 2—Supplementary Table 4, Supplementary Fig. 6—Supplementary Fig. 23). Notably, staple-bond crosslinking resulted in a substantial modification of the peptide NMR spectra. There were significant differences in $^{13}C_\alpha$ chemical shifts between stapled peptides **57/58** and the counterpart **59** (Fig. 4c). The temperature dependence of the amide NH groups was also significantly affected by the staple as dδ/dT values of peptides **57/58** are generally smaller than those of **59** (Fig. 4d). Some NH groups (IIe, Leu, His) of peptide **58** featured values characteristic of solvent-shielding ($<-3$ ppb/K) and the temperature coefficients for most NH groups of peptide **57/58** were reduced to the range of 2–4 ppb/K, indicating their involvement in wide-spread hydrogen bonds[61,62]. Several non-sequential NOEs were observed on peptide **57** and **58**, but not

peptide **59** (Fig. 4d, Supplementary Figs. 11, 17, and 23), indicating these stapled peptides were constrained by intramolecular cyclization to adopt more folded structures that are consistent with predictions from molecular modelling (Supplementary Fig. 24). These set of NMR characterizations confirmed the enhanced folding of peptides **57** and **58** which were FTDR-stapled at $X_L, X_L$ and $X_L, X_D$, respectively, compared to peptide **59** which was crosslinked at $X_D, X_L$ substrate, consistent with CD observations. The significant differences in NMR spectra shown for **57**, **58**, and **59** also suggested that their X substrate possessed different chirality; and peptides stapled by FTDR conditions did not result in strong epimerization. To further confirm this, we were able to show that these peptides eluted at different retention times by chiral HPLC separation (Supplementary Fig. 25).

To explore if the L,D versus D,L substrate preference for FTDR-based stapling is generally applicable to different peptide sequences, we turned our attention to another model peptide, which binds to the C-terminal region of an HIV-1 capsid assembly polyprotein (HIV C-CA) that is key to viral assembly and core condensation[63]. The 12-mer long peptide was previously demonstrated to have efficiently stabilized alpha-helical conformation once RCM-based stapling was performed at the indicated $i$ and $i + 4$ positions (**63**, Fig. 5)[26,64]. Using the optimized linker (1,3-benzenedimethanethiol), we performed FTDR-based stapling on the same sequence positions, and obtained analogues **60–62** that have fluoroacetamide substrates of different chirality combinations (Fig. 5a, Supplementary Table 5). FITC labelling was applied to the N-terminal during solid-phase synthesis in order to facilitate follow-up biological characterizations. Notably, stapling of the unprotected FITC-labelled peptides **60–62** proceeded smoothly despite the presence of FITC's thiourea moiety. As shown in CD spectra (Fig. 5b) and Table 1, stapling with L,L or L,D substrates generated peptides **60** and **61** that possessed similar helicity to the control peptide stapled by RCM. On the contrary, the crosslinked D,L substrate-containing analogue **62** only displayed a minimal level of helicity.

Taken together, FTDR-based stapling works most efficiently with the rigid meta-benzenemethane dithiol linker and generally

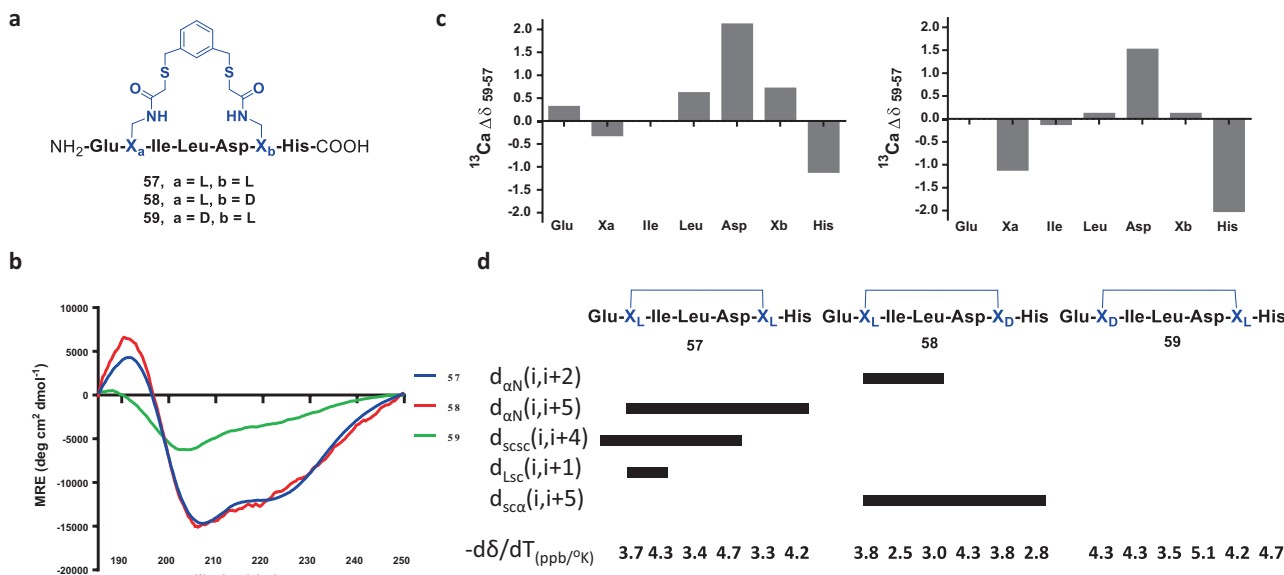

**Fig. 4 Peptide NMR spectra comparison among peptides 57–59 that are shortened representatives of the best FTDR-stapled Axin peptide analogues (39, 51, 55). a** Chemical structures of peptides **57–59**. **b** Circular dichroism (CD) spectra of **57–59**. **c** Plots of the $^{13}C_\alpha$ chemical shift differences ($^{13}C_\alpha$ $\Delta\delta_{58-57}$) between peptide **58** and **57**, and $^{13}C_\alpha$ chemical shift differences ($^{13}C_\alpha$ $\Delta\delta_{59-57}$) between peptide **59** and **57**, respectively. **d** Summary of the observed nonsequential NOE connectivity and temperature coefficients of the NH amide protons ($\Delta\delta/\Delta T$) of peptides **57–59**. "α": proton on the α carbon; "N": proton on the amide bond; "sc": proton on the side chain; "L": proton on the staple linker. Coupling residues $X_L/X_D$ and 1,3-benzenedimethanethiol linker are highlighted in blue.

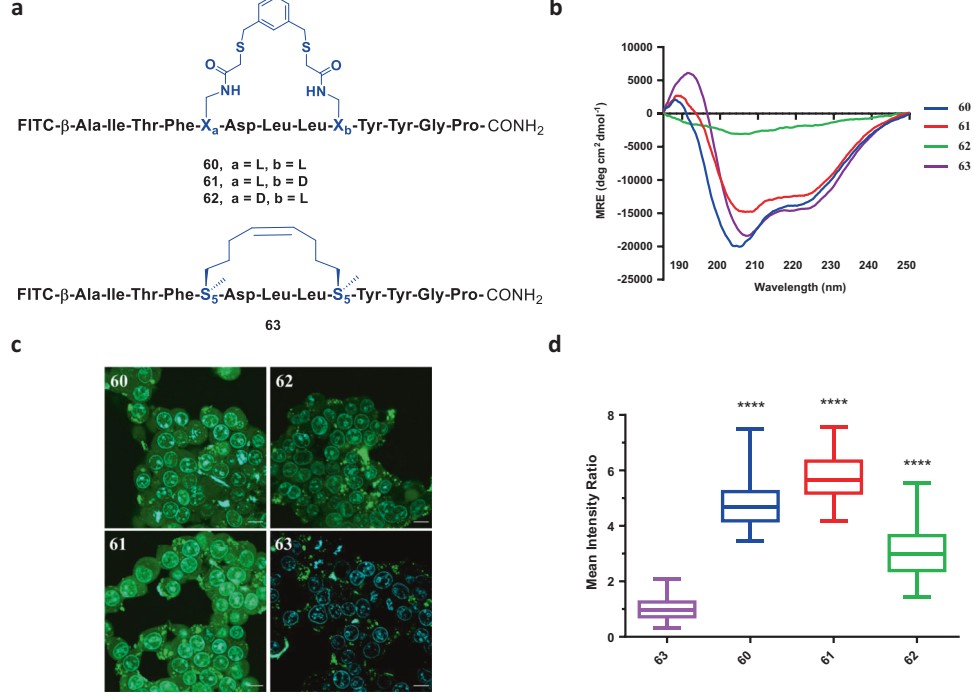

**Fig. 5 FTDR-stapled HIV C-CA binding peptide analogues. a** Chemical structures of peptides **60–62**. The reported HIV C-CA binding peptide **63** (stapled by RCM) was prepared as well. Stapling residues $X_L/X_D/S_5$ and the related linkers are highlighted in blue. **b** CD spectra of the crosslinked peptide analogues. **c** Fluorescent confocal microscopy images of the HEK293T cells treated with peptides **60–63**. Blue: nucleus stained by Hoechst 33342; Green: FITC-labelled peptides; Scale bar: 5 μm. **d** Quantification analysis of the cell penetration of peptides **60–63**. The mean intracellular intensity of peptide **63** was normalized as 1. Box plots show 25–75% quantiles, median values (bullseye), and minimal and maximal values (whiskers). Two-tailed unpaired t test with Welch's correction has been applied. "****" Represents $p < 0.0001$. Detailed sample sizes and mean ± SEM values are available at Supplementary Table 7.

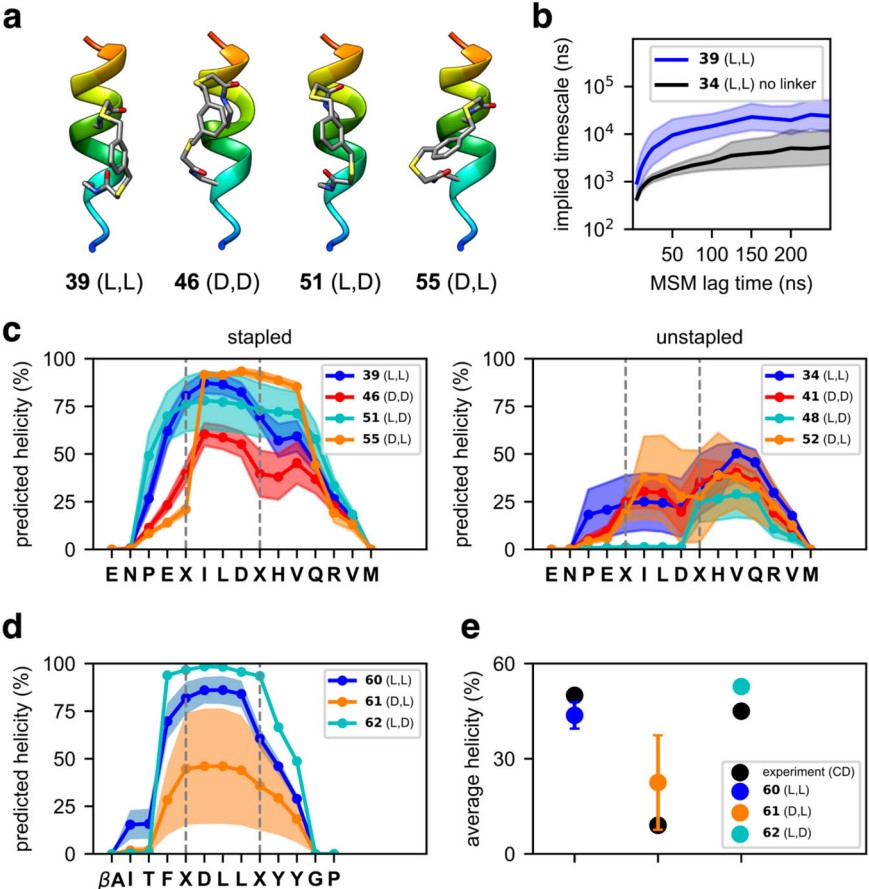

**Fig. 6 Molecular dynamics simulation. a** Energy-minimized structures of 1,3-benzenedimethanethiol crosslinked Axin peptides **39**, **46**, **51** and **55**. **b** The slowest implied timescale of peptides **39** and **34** (no linker) shown as a function of MSM lag time. **c** Predicted per-residue helicity profiles of stapled and unstapled Axin peptides. **d** Predicted per-residue helicity profiles of HIV C-CA binding peptides **60–62**. **e** Comparison of experimental (CD) and predicted average helicities of peptides **60–62**. Data in panels **b**–**e** are presented as mean values (dark lines) ± SEM (shaded regions) estimated from a bootstrap procedure where five MSMs ($n = 5$) were constructed by sampling the input trajectory data with replacement. .

prefers the L,L or L,D fluoroacetamide substrates on different peptide targets, with the D,L substrates least tolerated. To gain insight into the molecular mechanisms driving this preference, we performed extensive molecular dynamics simulations for stapled Axin peptides **35–55** and HIV C-CA-binding peptides **60–62**, generating an aggregate of 1.8 ms of trajectory data (Supplementary Table 6) on the Folding@home distributed computing platform[65–67]. Markov State Models[68,69] of the conformational dynamics (see the "Methods" section) showed that peptide stapling stabilizes helical conformations (Fig. 6a, c), and slows folding by an order of magnitude to the ~10 μs time scale (Fig. 6b). Predicted helicity profiles for stapled and unstapled Axin peptides (Fig. 6c, Supplementary Fig. 29) and stapled HIV C-CA binding peptides (Fig. 6d) also showed that crosslinked D,L substrate-containing peptide analogues disrupt helicity at the D-amino acid position more than those crosslinked L,D substrate-containing peptides. This is perhaps due to subtle differences in linker strain, combined with a differential helix nucleation propensities in the N- vs. C-terminal direction for stapled peptides[70]. In summary, predicted helicities from simulations compare well with experimental values from CD spectroscopy (Fig. 6e, Supplementary Figs. 30 and 31).

**Cell permeability of *i*, *i* + 4 FTDR-stapled peptide analogues.** We then turned our attention to the biological properties and activities of the stapled peptides. Cellular uptake of peptides has been revealed to be a complicated process driven by hydrophobicity,

positive charge, and alpha-helicity, etc.[35]. The RCM-based stapling was previously reported to endow enhanced cellular permeability to HIV C-CA binding peptide **63** due to the stabilized secondary structure and the increased hydrophobicity[64]. Particularly for stapled peptides, the staple type also serves as one of the deciding factors for cell penetration[36,37]. Thus, we wondered if peptides stapled by FTDR can render at least comparable cell permeability. We first investigated the dose-dependent cytotoxicity of analogues (**60–63**) after incubation with HEK293T cells for 12 h, and did not observe any significant effects on viability with up to 15 μM of peptides (Supplementary Fig. 32). Towards this end, all the FITC-labelled HIV C-CA binding analogues (**60–63**) at 10 μM concentration were incubated with HEK293T cells for 4 h and were subsequently analysed by confocal fluorescence microscopy (Fig. 5c). Significant cellular uptake of the FTDR-stapled peptides **60** and **61** were observed, with peptides not only spreading through the cytosol but also existing in the endosomes showing punctated greenish fluorescence. In comparison, much less fluorescence was observed from cells treated with the RCM stapled control **63**. A similar trend has been previously reported with HIV C-CA binding peptides stapled by perfluoroarylation of cysteines[26], suggesting that the aromatic component of the linkers may enhance the cellular uptake. Additionally, the weaker uptake of the D,L-stapled peptide (**62**) also corroborated the previously reported positive correlation between helicity and cell permeability[57].

Next, we asked if FTDR-based stapling on other peptide sequences can lead to enhanced cell penetration as well. To

observe the broader applications of FTDR-based stapling on other peptides and cell lines, we chose another Axin-derived peptide analogue (**68**, Fig. 7a), which displayed single-digit nM $K_d$ after $i$, $i + 4$ stapling by RCM, but showed limited cell permeability to DLD-1 cells[14]. Previously, other amino acids in this sequence had to be mutated into arginine in order to increase the overall positive charge. The resulting lead analogue had improved cellular uptake at the expense of losing 6–7 fold affinity towards the protein target β-catenin[14]. We thereby synthesized the FITC labelled fluoroacetamide containing L,L substrate (**64**) and L,D substrate (**66**), and decided to focus on the DLD-1 cell line as reported in order to facilitate direct comparison. FTDR-based stapling with 1,3-benzenedimethanethiol proceeded smoothly, and the resulting analogues (**65**, **67**, Fig. 7a, Supplementary Table 5) did not affect the viability of DLD-1 cells at 10 μM concentration after 12 h incubation (Supplementary Fig. 32). With confidence that there is no cytotoxicity, we treated DLD-1 cells with each of these peptides, along with the unstapled or RCM-stapled control peptides in parallel. As shown in Fig. 7b and Supplementary Fig. 33, significant cellular uptake was observed in cells incubated with 10 μM FTDR stapled **65** or **67**, while there was much weaker fluorescence in other control treatment groups including the cells incubated with **68**. The uptake pattern of **65** and **67** were similar to those observed earlier for HIV C-CA binding peptides **60** and **61**. A quantitative analysis was achieved by applying the lognormal fitting to a histogram of the individual cell mean intensity (Supplementary Figs. 34 and 35), and the resulting mean cellular fluorescence revealed that the FTDR stapled L,L HIV C-CA binding peptide **60** had $4.76 \pm 0.09$ fold of mean intensity compared to that of the RCM control peptide intracellularly, and the stapled L,D mimetic **61** had $5.75 \pm 0.07$ fold mean intensity (Fig. 5d, Supplementary Table 7). Quite consistently, the L,L Axin peptide derivative **65** showed a $4.86 \pm 0.15$ fold mean intensity compared to the RCM control, while the L,D Axin peptide mimetic **67** showed a $5.05 \pm 0.14$ fold increase (Fig. 7c, Supplementary Table 8). Moreover, the diminished uptake from unstapled peptides (e.g., $0.36 \pm 0.02$ fold for **64**, $0.33 \pm 0.01$ fold for **66**) further suggests the FTDR stapling is an essential requirement for cell permeability.

With the quantitative analysis tool in hand, we also applied the FTDR-based stapling to two other well-known peptide substrates (Stabilized alpha-helix of BCL-2 domains (SAHB_A)[13] and A-kinase targeting Stapled Anchoring Disruptors (STAD)[71]) that have been previously stapled by RCM at $i$, $i + 4$ sites. As shown in Supplementary Figs. 36a and 37a, FTDR-stapling of these long peptides resulted in analogues of higher macrocyclization yields than the RCM-stapled ones (Supplementary Table 5). Both types of staples promoted the α-helical folding to similar extents (Supplementary Figs. 36b, 37b, Table 1) and led to peptide penetration into mammalian cell lines without compromising cell viability (Supplementary Figs, 36 and 37). Nevertheless, SAHB_A with FTDR stapling on L,L (**70**) and L,D substrate (**72**) displayed $4.05 \pm 0.11$ fold and $2.41 \pm 0.09$ fold mean intensity compared to that of the RCM stapled peptide **73**, respectively (Supplementary Fig. 36e–f). STAD with FTDR stapling on L,L (**75**) and L,D substrate (**77**) also showed $2.92 \pm 0.08$ fold and $2.00 \pm 0.05$ fold mean intensity versus the RCM stapled STAD **78**, respectively (Supplementary Fig. 37d, e). Taken together, these results further supported that FTDR stapling more universally promoted cellular uptake than the commonly used RCM stapling strategy.

Despite intensive studies, the uptake mechanisms for cell-penetrating peptides remain ambiguous and largely varied due to their complicated nature[72–74]. Small molecule inhibitors specific for each endocytic pathway have been routinely utilized to interrogate the cellular uptake mechanisms of many transporters[75–78]. For example, nystatin as a sterol-binding agent was used to selectively block caveolin-dependent endocytosis[78], while chlorpromazine can selectively inhibit clathrin-mediated endocytosis[76,78]. Cytochalasin D would specifically induce depolymerization of actin and cease the subsequent apical endocytosis[77]. Sodium chlorate, on the other hand, aborts the decoration of cell membranes with sulfated proteoglycans, affecting certain other uptake pathways[75]. With those, RCM-stapled peptides were recently revealed to penetrate cells via clathrin- and caveolin-independent pathways that were partially mediated by cell surface proteoglycans[36]. To understand the mechanisms behind the enhanced cellular uptake of peptides stapled by FTDR, we performed similar experimental investigations using the lead Axin peptide derivatives **65** and **67**. The cell penetration experiments were repeated under conditions that blocked a different endocytotic pathway each round (Supplementary Figs 38 and 39). Cell viabilities were measured immediately after the imaging experiments to ensure that the observed results were due to the active cellular uptake (Supplementary Fig. 40). As quantitatively summarized in Fig. 7d and Supplementary Tables 9, 10, both peptides had their uptake partially blocked by more than one pathway-specific inhibitors. Like RCM-stapled peptides, both FTDR-stapled peptides internalized partially through sulfated proteoglycans, as indicated by the reduced uptake in cells treated by sodium chlorate. Yet unlike hydrocarbon-stapled peptides, the L,L stapled analogue **65** also partially penetrated cells via endocytosis depending on clathrin (inhibited by chlorpromazine) and actin polymerization (inhibited by cytochalasin D). The L,D stapled **67** appeared to enter cells additionally through clathrin- and caveolin-dependent (blocked by nystatin) endocytosis. Interestingly, the chirality difference in the $i + 4$ position between the stapled peptide **65** and **67** seemed to result in different peptide backbone conformations, thereby affecting their intake through distinguished endocytosis pathway. Taken together, our data suggested that FTDR-stapled peptides may penetrate cells through multiple endocytotic pathways in a distinct pattern compared to the RCM-stapled peptides, which could account for their enhanced cellular uptake than those observed for the RCM-stapled controls.

**Activity of $i$, $i + 4$ FTDR-stapled peptide analogues.** In addition to cell permeability, we wondered whether the structural features brought by the 1,3-benzenedimethanethiol crosslinker translates to other functional relevance. Thus, we performed ELISA assay to quantify the binding affinity of the lead Axin peptide derivatives towards the target protein β-catenin (Fig. 7e). The $EC_{50}$ was $4.36 \pm 1.75$ nM for stapled analogue **65**, and $5.27 \pm 2.29$ nM for analogue **67**, which were similar to the $EC_{50}$ of RCM-stapled **68** ($3.03 \pm 1.8$ nM) and were at least 100-fold more potent than unstapled peptide **64**. Given that staples and the peptide conformational changes after stapling usually render the structures less prone to protease-mediated cleavage[15,18,26], we also tested the serum stability of these lead peptides in 100% rat serum. As shown in Fig. 7f, all the stapled peptides remained mostly intact while unstapled control **64** was rapidly degraded, with only 30.9% left after 20 min of incubation. In order to see if their improved cellular uptake can translate to enhanced cellular activity, we finally examined these analogues' inhibition of the growth of a Wnt-driven colorectal cancer cell line[14] DLD-1 over a 5-day period (Fig. 7g). FTDR stapled **65** and **67** potently impeded the cell growth with $EC_{50}$ values of $2.3 \pm 0.4$ and $9.8 \pm 2.0$ μM, respectively. On the contrary, neither the unstapled control **64** nor the RCM-stapled analogue **68** displayed a significant growth inhibition until they were administered at 16 μM concentration. Given that these peptide analogues did not display cytotoxicity within 12 h of incubation (Supplementary Fig. 32), we also

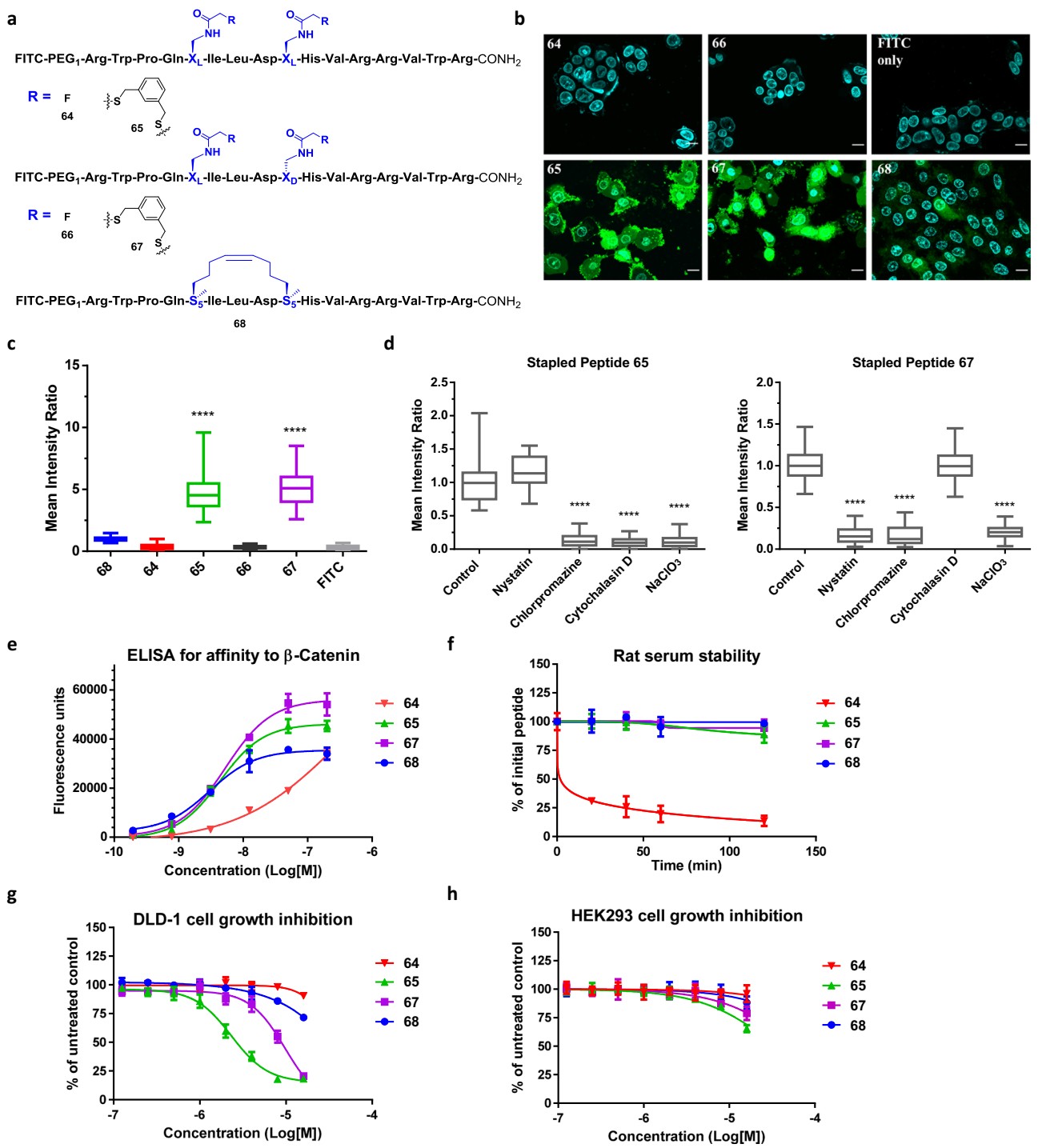

**Fig. 7 FTDR-stapled functional Axin peptide analogues. a** Chemical structures of peptides **64**–**67**. The Axin peptide analogue **68** (stapled by RCM and reported for cell penetration studies) was prepared as well. Stapling residues $X_L/X_D/S_5$ are highlighted in blue. **b** Fluorescent confocal microscopy images of the DLD-1 cells treated with peptides **64**–**68** and FITC only as a negative control. Blue: nucleus stained by Hoechst 33342; Green: FITC; Scale bar: 5 µm. **c** Quantification analysis of the cell penetration of peptides **64**–**68**. The mean intracellular intensity of peptide **68** was normalized as 1. "****" Represents $p < 0.0001$. **d** Quantification analysis of the cell penetration of peptide **65** or **67** after the cells had been treated with blockers of endocytic pathways. The mean intracellular intensity of peptides in cells untreated with any blocker was normalized as 1. See SI for fluorescent microscopy images. "****" Represents $p < 0.0001$. For **c** and **d**, box plots show 25–75% quantiles, median values (bullseye), and minimal and maximal values (whiskers). Two-sided unpaired $t$ test with Welch's correction has been applied. Detailed sample sizes, mean ± SEM values are available at Supplementary Tables 8–10. **e** The binding affinity of unstapled (**64**) and stapled Axin peptide analogues (**65**–**68**) to the target β-catenin protein. **f** The stability of the aforementioned peptide analogues in 100% rat serum. **g**–**h** The inhibition of these peptide analogues on the 5-day growth of DLD-1 cancer cells (**g**) and HEK293 control cells (**h**). Data plots in panels **e**-**h** are mean values ± standard deviation (SD) measured from $n = 3$ biologically independent samples.

titrated their effects on DLD-1 cells over a time course of 24–96 h (Supplementary Fig. 41), and only observed significant effects from peptides **65** and **67** after 48 h of incubation. This suggests that the FTDR-stapled Axin peptide leads possessed minimal cytotoxicity but greater growth inhibition properties. Following this phenotypic observation, we also performed the TOPFlash luciferase reporter assay in DLD-1 cells and detected strong inhibition of β-catenin mediated transcription only by peptides **65** and **67** (Supplementary Fig. 42a). Lastly, we tested these peptides' phenotypic activity in the HEK293 cell line which has only a basal level of Wnt signalling (much lower expression of endogenous β-catenin than DLD-1 cells)[79,80]. As shown in Fig. 7h, little effects were observed for peptide **65** and **67** on the growth of HEK293 control cells. These results further demonstrated that these FTDR-stapled peptides specifically work on Axin/β-catenin-mediated Wnt signalling pathway.

**Exploration of FTDR stapling at $i$, $i + 7$ positions**. The promising results of the $i$, $i + 4$ stapling prompted us to test the application of the FTDR reaction to staple peptides at $i$, $i + 7$ positions, which have been previously demonstrated by RCM on a 16-residue α-helical transactivation domain of p53[81,82]. Thus, we prepared the corresponding peptide analogues incorporating the fluoroacetamide building blocks with L,L (**79**) or L,D (**84**) configurations at the reported $i$, $i + 7$ positions (Fig. 8a). Initial macrocyclization studies with the 1,3-benzenedimethanethiol linker surprisingly did not result in the desired cyclization product at a detectable level but mostly the mono- or di- substituted side products. Cyclized products can be only achieved via reactions with aliphatic linkers equal to or longer than 7 carbon or para-positioned phenyl linkers (Fig. 8a, Supplementary Table 5). Cyclization with 1,4-benzenedimethanethiol at $i$, $i + 7$ positions appeared to render the highest yield (>70%) after HPLC purification (Supplementary Table 5). CD studies (Fig. 8b, Table 1) further revealed that the 1,4-benzenedimethanethiol-crosslinked p53 peptide analogues (**82**, **87**) in either L,L or L,D configurations retained the most stabilized alpha helical structures, similar to the RCM-stapled control analogue **89**.

To evaluate if the enhanced helical folding of p53 peptide mimetics translates to improved biological activities, we first tested their competition with the formation of wild type p53–HDM2 protein complexes by an ELISA assay (Fig. 8c). The FTDR-stapled analogues **82** and **87** displayed similar potencies ($EC_{50}$ ~ 100 nM) to the RCM-stapled p53 peptide **89**, which has been reported to have $EC_{50}$ values between 100 and 200 nM in HDM2 competition binding assays that is consistent with our results[23,81]. On the contrary, the unstapled peptide **79** only displayed modest competition starting from 0.8 μM. We then prepared the N-terminal FITC-labelled versions (**90–94**, Fig. 8a) and studied their cellular uptake in p53 wild-type HCT-116 colorectal carcinoma cells which were reported to respond to p53 peptides[23,81]. As shown in Fig. 8d, greatly enhanced cell penetration was observed for the FTDR-stapled peptides after 4 h of incubation (**91**, **93**). Analogues **91** and **93** displayed 7.9 ± 0.09 fold and 8.5 ± 0.22 fold mean intensity, respectively, compared to the RCM control peptide **94** (Fig. 8e). The observed minimum staining of cells by propidium iodide suggested the integrity of cell membranes after 4 h treatment with our peptides at a final concentration of 2 μM (Fig. 8d). Finally, we explored these peptide analogues' cytotoxicity in p53 wild-type HCT-116 cell line, which has been reported to undergo apoptosis induced by RCM-stapled p53 peptide after 24 h of treatment[23,81]. We were able to observe similar apoptotic activity from our FTDR-stapled peptides (Fig. 8f). Specifically, analogues **82** and **87** have an $EC_{50}$ of 0.66 ± 0.19 μM and 0.71 ± 0.18 μM, respectively, slightly better

than the RCM control **89** which has an $EC_{50}$ of 1.1 ± 0.33 μM. On the contrary, the unstapled peptide **79** was inactive for all concentrations below 10 μM. These series of peptides also demonstrated their selectivity with minimal levels of activity in the p53-null HCT-116 cell line (Fig. 8g). Lastly, we also validated the specificity of peptides **82** and **87** towards the p53-mediated pathway using the PG13 luciferase reporter assay (Supplementary Fig. 42b). Collectively, FTDR-stapling based on the $i$, $i + 7$ linkage has resulted in a class of peptide analogues similar in structure and biological activity to the RCM-stapled peptides. Consistent with the results on the $i$, $i + 4$ linkage, the FTDR-stapled peptides displayed enhanced cellular uptake in comparison to RCM-stapled peptides.

## Discussion

We have demonstrated a new, mild, and clean synthetic strategy to cyclize and/or staple unprotected peptides. The developed fluorine–thiol displacement reaction (FTDR) approach operates at mild temperature in aqueous solutions and offers excellent chemoselectivity and functional group tolerance, e.g., sparing intrinsic cysteines. We first exemplified its application as a general macrocyclization platform that can be compatible with a variety of linkers. Then we demonstrated its use in stapling peptides at both $i$, $i + 4$ positions and $i$, $i + 7$ positions, and further showed that the lead stapled peptides retained the structure features and biological properties reported in literature. The identification of benzenedimethanethiol derivatives as the optimal linkers for various stapling applications suggests that certain aromatic rigidity in the crosslinker region is required to maintain the alpha-helical conformation of peptide substrates stapled by FTDR. The subtle change of dimethanethiol positions from meta (1,3-disubstituted) to para-(1,4-disubstituted) on benzene expanded the substrate scope of stapling from $i$, $i + 4$ to $i$, $i + 7$, showcasing the fine-tuning of chemistry to affect peptide folding. Furthermore, both the experimental results and the molecular dynamics simulation consistently pinpointed the preference of L,L and L,D substrate chirality for the folding (alpha helicity) of the FTDR-stapled peptides, implicating the distinct helix nucleation propensities in the N- vs. -C-terminal direction for this class of stapled peptides.

In terms of biological functions, the enhanced cellular uptake of stapled peptides and the associated distinct penetration mechanism as evidenced by the lead Axin peptide analogues suggest this FTDR-based stapling approach may expand the toolbox of chemical transformations to generate a new class of probes or therapeutic leads for intracellular targets. To the best of our knowledge, there have been few reported efforts to elucidate the cellular uptake mechanism of peptides stapled by strategies other than RCM. During the manuscript review and revisions, several peptidomimetics have been also revealed to enter cancer cells via micropinocytosis[83]. Our findings confirmed the previous observations that there could be more than one uptake mechanisms existing for stapled peptides[36]. Further, they were consistent with the previous discovery that internalization of stapled peptides mainly correlated with the staple type[36]. Accordingly, our approach in many aspects is complementary to the RCM-mediated peptide stapling. Current efforts are focused on further optimization of FTDR reaction conditions with the exploitation of other nucleophilic crosslinkers. Application of the FTDR-based stapling to probe PPIs related to the key signalling events in prostate cancer and neurodegenerative diseases are also under active investigation in our research group.

In summary, unprotected peptides stapled by FTDR appeared to recapitulate the advantages in biological functions seen in the classic RCM-stapled peptides, but also possessed enhanced cell

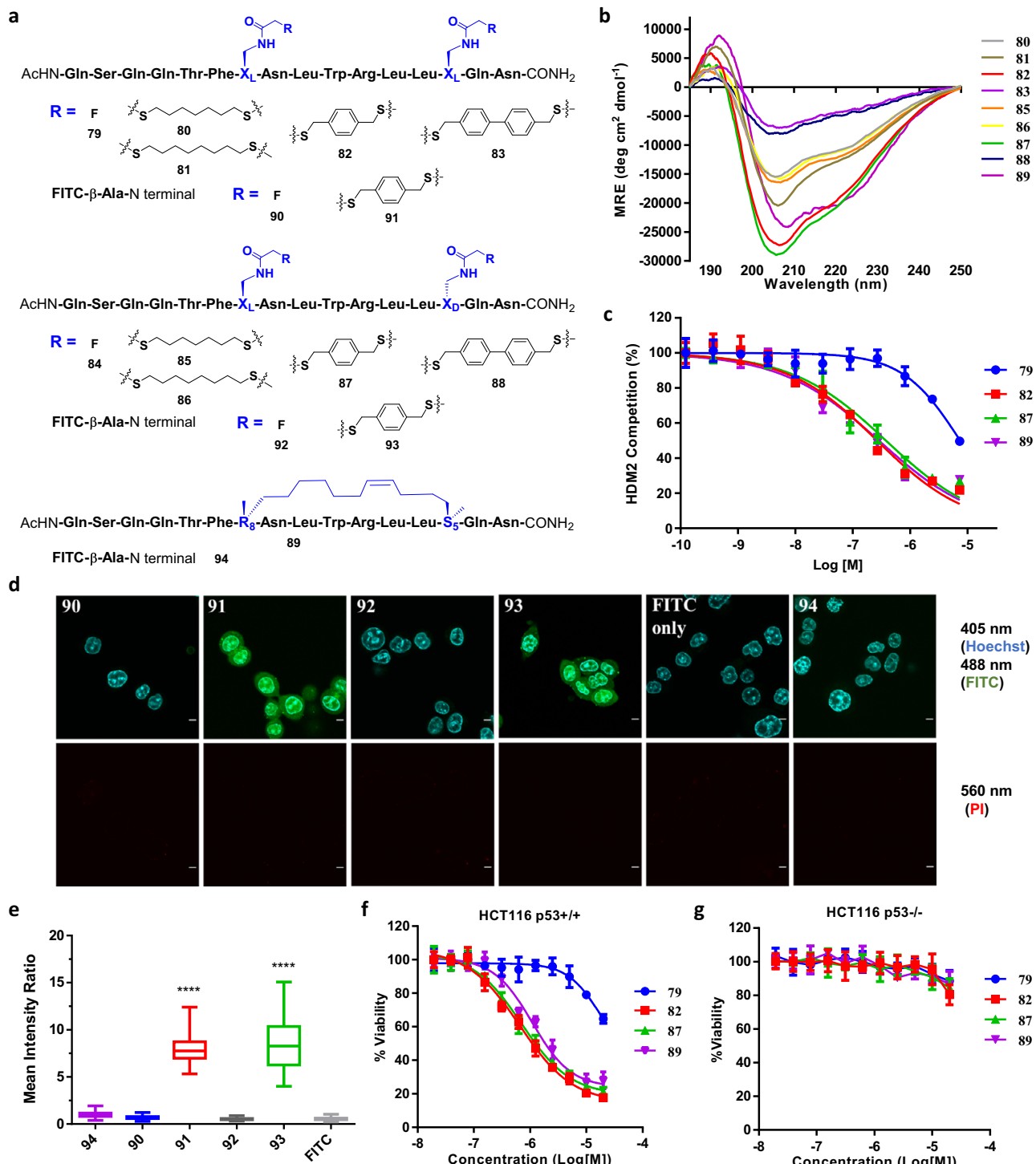

**Fig. 8 FTDR-stapled p53 peptide analogues. a** FTDR coupling between unprotected p53 peptide analogues (**79**, **84**) and various dithiol linkers with $i$, $i + 7$ linkage. The reported p53 peptide analogue **89** was prepared based on RCM stapling. Coupling residues $X_L/X_D/R_8/S_5$ are highlighted in blue. Peptides **90**–**94** are the N-terminal FITC labelled versions of peptide **79**, **82**, **84**, **87**, and **89**. **b** CD spectra of all crosslinked p53 peptide analogues. **c** ELISA assay of stapled p53 peptide analogues (**82**, **87**, RCM control **89**) and the unstapled control **79** for their inhibition of p53-HDM2 interaction. Data represents mean values ± SD ($n = 3$ independent samples). **d** Fluorescent confocal microscopy images of the wild type HCT116 cells treated with the N-terminal FITC labelled peptides **90**–**94** and FITC as a negative control. Blue (405 nm channel): nucleus stained by Hoechst 33342; Green (488 nm channel): FITC; Red (560 nm channel): propidium iodide staining; Scale bar: 5 μm. **e** Quantification analysis of the cell penetration of peptides **90**–**94** ($n = 159$ for **94**, $n = 144$ for **90**, $n = 163$ for **91**, $n = 134$ for **92**, $n = 155$ for **93**, $n = 148$ for FITC only). The mean intracellular intensity of the RCM-stapled peptide **94** was normalized as 1. Box plots show 25–75% quantiles, median values (bullseye), and minimal and maximal values (whiskers). Two-sided unpaired $t$ test with Welch's correction has been applied. "****" Represents $p < 0.0001$. **f**–**g** Cell viability assays in p53 wild-type HCT-116 cells (**f**) and p53 null HCT-116 cells (**g**) treated with well-stapled p53 peptide analogues. Data plots are mean values ± SD based on $n = 3$ independent samples.

permeability and the correlated growth inhibition of the targeted cancer cells.

## Methods

Chemical synthesis procedures, supporting tables, and supporting figures are available as Supplementary Information.

**Peptides synthesis**. Peptides were synthesized on rink amide resins following traditional Fmoc-based solid-phase chemistry. Most >10 mer peptides were synthesized on an automatic peptide synthesizer at the Macromolecular core facility of the Pennsylvania State University. Generally, approximately four equivalents of Fmoc-protected amino acid building blocks (including compound **15** for $X_L$ or compound **16** for $X_D$), four equivalents of HATU, and eight equivalents of DIPEA were added to the resins resuspended in DMF for every round of coupling, which has been reported to minimize epimerization[84]. The N-terminal of peptides were either left uncapped or capped with acetic anhydride or FITC in DMF using DIPEA as the base. Final peptides were cleaved by incubating the beads with reagent H (trifluoroacetic acid (81% w/w), phenol (5% w/w), thioanisole (5% w/w), 1,2-ethanedithiol (2.5% w/w), dimethylsulfide (2% w/w), ammonium iodide (1.5% w/w), and water (3% w/w))[85] at room temperature for 3 h. The supernatant was collected and added diethyl ether to precipitate out the crude product. The crude mixture was re-dissolved in a water and methanol mix, and purified by reverse phase semi-preparative HPLC that operated at a flow rate of 10 mL/min, used water/0.1% TFA as solvent A and acetonitrile/0.1% TFA as solvent B.

**General procedures for peptide stapling**. For the controls that require ring-closing metathesis (RCM), RCM-based stapling was performed with protected peptides on resins following the reported procedures[14]. The resins were mixed with ~0.5 equivalent of Grubbs I catalyst (5 mM) in dichloromethane and were incubated for 2 h at room temperature before the solvents were drained off. The coupling process was repeated three times, followed by subsequent N-terminal deprotection and capping. The crude products were cleaved from resins, precipitated out by 15-fold ice-cold diethyl ether, and purified by HPLC as mentioned above.

For stapling based on the fluorine thiol displacement reaction (FTDR), 252 μL of the dithiol-containing linker (250 mM in DMF) was premixed with 420 μL of sodium hydroxide solution (250 mM) and 378 μL of DMF at room temperature for 40 min to completely deprotonate dithiols. After this, 105 μL of peptide solution (50 mM in water or DMF) was added to start the stapling based on FTDR. For peptides containing cysteines, 2.5 equivalent of TCEP in aqueous solution (pH 7.0) was premixed with the peptide solution to retain cysteine in a reduced reactive status. The reaction mixture of dithiol linkers and peptides has a final pH of 9.0–9.5 and was incubated at 37 °C for around 4–12 h till the reaction is complete as determined by LC–MS, and was then quenched with water and acetic acid. Subsequently, the solution was extracted by ethyl acetate three times to remove excess linkers. The remaining aqueous phase was lyophilized, resuspended in methanol, and added diethyl ether to precipitate out the stapled peptides. The crude product was further purified on HPLC as mentioned before. The macrocyclization yields for all the stapled peptides (Supplementary Tables 1 and 5) were determined after recovery from HPLC purification.

**Circular dichroism**. Axin- and p53-derived peptides were dissolved in water, and HIV-targeting peptides were dissolved in 25% acetonitrile/water to ensure 100% solubility. The final concentration of the peptide samples was 70 μM. Circular dichroism (CD) measurements were performed on a Jasco model J-815 spectropolarimeter with a 1 mm Jasco quartz cell over the wavelength range of 180–250 nm. The scans were carried out at 0.2 nm resolution with 4 s average time at 25 °C. Data from three scans were averaged, base line corrected, and normalized to the mean residue ellipticity (MRE) following the equation: $[\theta]_\lambda = [\theta]_{obs}/(10 \times l \times C \times n)$. $[\theta]_\lambda$ is MRE in $deg \times cm^2 \times dmol^{-1}$; $[\theta]_{obs}$ is the measured ellipticity; $C$ is the concentration of peptides in M; $l$ is the optical pathlength in cm; and $n$ is the number of residues in peptides. The % alpha helicity was calculated from the MRE values at 222 nm, using the equation % helicity = $([\theta]_{222}-[\theta]_0)/([\theta]_{max}-[\theta]_0)$ based on the previously reported method[86]. $[\theta]_{222}$ is the MRE value at 222 nm; $[\theta]_{max}$ is the maximum theoretical MRE value for a helix of n residues; and $[\theta]_0$ is the MRE value of the peptide in random coil conformation that usually equals to (2220-53T)[86].

**NMR analysis of peptides**. NMR spectra of peptides were obtained in DMSO-d6 using Shigemi tubes with a Bruker AV-III 500 spectrometer. The data were collected by Bruker TopSpin software and processed by MestreNova Version 14.2. $^1H$ resonances were analysed using two-dimensional NMR experiments (COSY, HMBC and ROESY). APT $^{13}C$ spectra were recorded for peptides where primary and tertiary carbons appear as negative peaks; secondary and quaternary carbons appear as positive peaks. $^1H$–$^{13}C$ HSQC spectra were utilized to assign $^{13}C$ resonances based on the observed cross-correlations. The residual DMSO signals ($^1H$, 2.49 ppm; $^{13}C$, 39.5 ppm) were used as references for the spectra. Mixing time in all the NMR experiments was 200 ms. The temperature coefficients were determined via $^1H$ spectra in the 293–373 K range with a step size of 20 K for the amide protons.

**Cell viability assay for imaging studies**. Wild type DLD-1 cells were cultured in complete growth media (RPMI, 10% FBS and 1% penicillin/streptomycin) at 37 °C/5% CO₂. After harvesting, cells were placed onto 96-well flat bottom white plates at 10e4 cells/well, and were incubated overnight. The next morning, serially diluted axin analogues (final concentration 0, 5, 10, and 15 μM) were added to the cell culture media and incubated for 12 h. The cells were then treated with CellTiter-Glo® reagent that had been prepared following the manufacturer's protocol at 100 μL/well. The chemiluminescent signals were collected on a Synergy H1 plate reader (Biotek) using Gen5 software, and the signals from the control sample (treated with 0 μM peptides) were used as the 100% control. For HIV-1 peptide analogues, the samples were incubated with HEK293T cells instead, and the rest of the procedure was the same as mentioned before. For Jurkat cells, after imaging studies with SAHB$_A$ analogues (5 μM, 4 h of incubation), viabilities were measured by mixing with the CellTiter-Glo reagent and the chemiluminescence signals were recorded and processed the same as mentioned above. Similarly, confocal imaging of DLD-1 cells for the penetration pathway studies were followed by viability measurements. Cells treated with vehicle groups were also analysed and counted as the 100% control.

**Confocal imaging**. All images were captured on an Olympus FV3000 confocal laser scanning microscope with a NA 1.05 ×30 silicone-immersion objective (UPLSAPO 30XS). Hoechst 33342 dye and FITC-labelled peptides were excited with 405 and 488 nm lasers (Coherent OBIS), respectively. Images were analysed with NIH ImageJ software. Using the "analyse particles" function, we measured the area and intensity of individual cells (Supplementary Fig. 34a)[87]. Particles in the extracellular matrix that the "analyse particle" function selected were manually deselected and removed from the data set (Supplementary Fig. 34b). Cells with cytoplasmic membrane aggregation of the FITC-labelled peptide had the cytoplasmic membrane excluded from the mean intensity measurement (Supplementary Fig. 34c). From a normal distribution curve applied to a histogram of the individual cell mean intensity, we found some outliers in the data collected (Supplementary Fig. 35a). However, from the distribution of data we found the best fit curve to be lognormal. Outliers were then removed using a Grubbs test (Supplementary Fig. 35b)[88]. We determined the optimal bin size for the histogram using the Freedman–Draconis rule[89]. The mean fluorescence intensity from each treatment group was calculated from $n \geq 100$ cells. Error bars represented SE. $P$ values were calculated by two sided Welch's $t$ test for comparison with internalization of RCM-stapled control peptides.

**Cell penetration assay**. DLD-1 cells were seeded on 35 mm optical dishes at 3e5 per well and were incubated overnight at 37 °C/5% CO₂. FITC-labelled Axin peptide analogues were added and incubated for 12 h at 10 μM final concentration[14]. To visualize the nucleus, the cells were incubated with Hoechst 33342 (1 μg/mL, PBS, ThermoFisher) for 10 min. Cells were imaged in buffer mimicking the physiological conditions in the cytoplasm (20 mM HEPES, 110 mM KOAc, 5 mM NaOAc, 2 mM MgOAc, 1 mM EGTA, pH 7.3). FITC-labelled HIV-1 C-CA peptides were imaged similarly, except that the incubation was with HEK293T cells for 4 h[26]. For FITC-labelled SAHB$_A$ analogues, 5 μM of each was incubated with Jurkat cells for 4 h[13]. The rest of the imaging procedures were done the same as mentioned above. Likewise, 5 μM of STAD peptides were incubated with HeLa cells for 2 h[71], followed by similar imaging studies; and 2 μM of p53 peptide analogues were mixed with HCT116 wt cells for 4 h before the imaging analysis of their cellular uptake. Cell samples treated with STAD or p53 series of peptide analogues were also simultaneously stained by propidium iodide (PI) to ensure the cell viability and membrane integrity.

**Cell penetration pathway study**. The plated DLD-1 cells were incubated for 1 h with 25 μg/mL nystatin for blocking caveolin-mediated endocytosis, 5 μg/mL chlorpromazine for blocking clathrin-dependent endocytosis, 10 μg/mL cytochalasin D for inhibiting actin polymerization, and 80 mM NaClO₃ for disrupting proteglycan synthesis according to the conditions used by literature[36]. After this, the FITC-labelled peptide analogues were added at a final concentration of 10 μM, and the mixture was incubated at 37 °C/5% CO₂ for 4 h. The cells were washed with PBS and incubated with Hoechst 33342 (1 μg/mL, PBS) for 10 min. After another round of washing, cells were imaged in the buffer as aforementioned. Subsequent CellTiter-Glo viability measurements were also performed directly after confocal imaging.

**ELISA assay for binding affinity**. The beta-catenin protein (Abcam, ab63175) (1 μg/mL, 50 μL/well) was coated onto a 96-well flat bottom black plate (Nunc, MaxiSorp) at room temperature for 2 h. The wells were washed with 0.05% tween containing PBS buffer, and then blocked with 1% BSA containing PBS buffer at room temperature for 2 h. The FITC labelled Axin peptide derivatives (including the unstapled control) were each serially diluted (200, 50, 12.5, 3.125, 0.781, 0.195 μM, and 0) in 100 μL of blocking solution (1% BSA, PBS buffer), added to the wells, and incubated at room temperature for another 2 h. The wells were then

washed three times with PBS buffer (0.05% tween) and treated with HRP-conjugated anti-FITC antibody (1:500 dilution, 100 μL/well, Abcam, ab196968) for 1 h at room temperature. After this final incubation, the wells were washed five times with 300 μL of PBS buffer (0.05% tween), and then treated with the QuantaBlu fluorogenic peroxidase substrates that had been prepared following the manufacturer's protocol (100 μL/well). The fluorescence signals were recorded by the H1 synergy plate reader at the excitation wavelength of 325 nm and the emission wavelength of 420 nm, using the Gen5 software.

**Serum stability assay.** Based on the ELISA-based assay of $EC_{50}$'s (~4 nM) for all the stapled peptides' binding with beta-catenin, FITC-labelled peptide samples ($L_i$, $_{i+4}L$, $L_i$, $_{i+4}D$ or the RCM control) were dissolved in 100% rat serum (Sigma Aldrich) at an initial concentration of 4 μM. Given the $EC_{50}$ value (~300 nM) of the unstapled peptide, a concentration of 300 μM was initially used for incubation with rat serum. Each sample mixture was then equally aliquoted into 15 tubes, and were incubated at 37 °C. At every specified time point (0, 20, 40, 60 and 120 min), 3 of the tubes were collected, and immediately diluted 1/1000 with PBS buffer (1% BSA), followed by flash freezing in liquid nitrogen and short-term storage at −80 °C. On the day of the ELISA assay, all the samples were slowly thawed on ice, and the remaining percentage of active peptides were determined by the ELISA assay following the procedure described above. The signals from the peptide sample mixture at 0 min were used as the 100% control.

**Cell cytotoxicity and growth inhibition assays.** For growth inhibition by Axin peptides, DLD-1 cells were cultured and maintained as mentioned before. Right before the assay, cells were resuspended in fresh RPMI media supplemented with 10% FBS, 100 IU/mL penicillin, and 100 μg/mL streptomycin, and plated onto 96-well white flat-bottom plates at 1000 cells per well (90 μL media/well). After overnight incubation, peptide samples were serially diluted as 10× stock in PBS buffer (10% DMSO), and were added to the plated cells at 10 μL stock solution/well in triplicate to make final treatment concentrations of 0, 0.0625, 0.125, 0.25, 0.5, 1, 2, 4, 8, 16 μM. The wells on the edges of plates were filled with PBS buffer to avoid unwanted evaporation of samples. The samples on the plates were incubated at 37 °C, 5% $CO_2$ for 5 days, and the cell viability was evaluated by CellTiter Glo (Promega). The luminescence signals were recorded by the H1 synergy plate reader (Biotek) using the Gen5 software, and were normalized against the control groups (100%) for which cells were only treated with the vehicle (PBS/1% DMSO). Control HEK293 cells were cultured in DMEM media supplemented with 10% FBS, 1% penicillin/streptomycin at 37 °C, 5% $CO_2$, then incubated with peptide samples and measured for viability in the same manner as mentioned above. For Axin peptide analogues' time-dependent cytotoxicity, peptide analogues were mixed with cells (1000 cells per well) at a final concentration of 0, 5, 10, and 15 μM. The cell viabilities we meaured via CellTiter Glo after incubation of 1 day, 2 days, and 4 days, respectively. For the cytotoxicity by p53 peptide analogues, HCT116 (p53+/+ or the control p53−/−) cells (a gift from Professor Bert Vogelstein, Johns Hopkins) were cultured in McCoy's 5A media (containing 10% FBS and 1% penicillin/streptomycin) at 37 °C/5% $CO_2$, plated at 1000 cells/well, and incubated with p53 peptide analogues at a final peptide concentration of 0, 0.039, 0.078, 0.125, 0.156, 0.313, 0.625, 1.25, 2.5, 5, 10, and 20 μM, respectively. After 24 h of incubation, the cell viabilities were measured and normalized against the vehicle control groups the same as mentioned above.

**Luciferase reporter assays.** For evaluation of Axin peptide analogues, DLD-1 cells were cultured in 96-well flat bottom plates up to 70–80% confluency and were transfected with 200 ng M50 TOPFlash plasmid (a gift from Professor Raymond Habas, Temple Biology) and 50 ng pRL-TK plasmid (Promega). 12 h after transfection, cells were treated with DMSO vehicle or 15 μM of Axin peptide analogues for another 24 h. Luciferase activity was measured using the Dual-Glo luciferase assay kit (Promega) on a Biotek Synergy H1 plate reader using the Gen5 software. Data were processed as firefly: Renilla luciferase ratio for each control or treatment group, and were further normalized against the vehicle control group (100%) as the Relative Activities.

For evaluation of P53 peptide analogues, HCT116 wt cells were transfected with 200 ng PG13-luc plasmid (a gift from Bert Vogelstein (Addgene plasmid #16442; http://n2t.net/addgene:16442; RRID:Addgene_16442)) and 50 ng pRL-TK plasmid. At about 12 h after transfection, the cells were treated with solvent vehicle or 15 μM of each p53 peptide analogue for an additional 24 h. The follow-up luciferase activity measurement and data processing were completed in the same manner as mentioned above.

**In vitro competition assay of p53–HDM2 interaction.** The p53–HDM2 ELISA kit was purchased from Enzo Life Sciences (catalogue #ADI-960-070) and the assay was performed following the manufacturer's protocol. Briefly, the stapled p53 peptide mimetics were serially diluted into assay buffer followed by the addition of wt p53 protein (12.6 ng/mL final concentration) and HDM2 protein (2.5 ng/mL final concentration). After 1 h of incubation at room temperature, 100 μL of the sample mixture was added to the 96-well flat bottom plate that had been previously coated with the p53 capture antibody (1:250 dilution) and blocked with BSA. After another hour of incubation at room temperature followed by washing, the

remaining samples in the wells were probed by 100 μL of the biotin conjugated HDM2 detection antibody (1:250 dilution). The wells were then washed with the wash buffer (10 mM sodium phosphate, 15 mM NaCl, 0.1% Tween-20, pH 7.4) and incubated with 100 μL of HRP-conjugated streptavidin. After this final incubation, the wells were extensively washed 5 times and treated with the QuantaBlu fluorogenic peroxidase substrates for fluorescence signals at ex 325 nm/em 420 nm.

**Molecular dynamics simulation.** Molecular dynamics (MD) simulations of stapled/unstapled Axin peptides and HIV peptides were performed using the OpenMM 7.0 simulation package[90] on the Folding@home distributed computing platform[91]. The AMBER ff14SB force field[92] was used for the peptide residues, while non-natural residues and linkers used GAFF[93] with partial charges from AM1-BCC[94], parameterized using the *antechamber* package of AmberTools17[95]. Axin peptide simulations were initiated from helical conformations taken from crystal structure PDB:1QZ7. These structures were solvated in a ~$(55\,\text{Å})^3$ cubic periodic box with TIP3P water molecules and $Na^+$ and $Cl^-$ counterions at 100 mM to neutralize charge, for a total of ~16 K atoms. HIV peptide simulations were initiated from helical conformations taken from the crystal structure PDB:3V3B. These structures were solvated in a ~$(45\,\text{Å})^3$ cubic periodic box with $Na^+$ and $Cl^-$ counterions, for a simulation size of around 9K atoms.

Trajectory production runs were performed using stochastic (Langevin) integration at 300 K with a 2-fs time step. Covalent hydrogen bond lengths were constrained using the LINCS algorithm. PME electrostatics were used with a nonbonded cutoff of 9 Å. The NVT ensemble was enforced using a Berendsen thermostat. About 50 trajectories were generated for each peptide design, reaching an average trajectory length of ~1.0 μs. In total, 1.8 ms of aggregate simulation data were generated for all designs, with about ~70 μs of trajectory data per design (Supplementary Table 6). Trajectory snapshots for the peptide coordinates were saved for every 500 ps. For all analysis described below, the first 100 ns was discarded from each trajectory to help remove systematic bias.

**Markov state model construction.** To describe the conformational dynamics of each peptide, Markov state models (MSMs) were constructed from the trajectory data using the MSMBuilder 3.8.0 software package[96]. This involved the following steps: (1) performing dimensionality reduction using time-structure-based independent component analysis (tICA)[97,98], (2) conformational clustering in the reduced space to define and assign the trajectory data to discrete metastable states, and (3) estimating the transition rates between metastable states from the observed transitions between states.

The tICA method is a popular approach to project protein coordinates to a low-dimensional subspace representing the degrees of freedom along which the slowest motions occur. In tICA, structural features $f_i$ are computed for each trajectory frame, and the time-lagged correlation matrix $C(\Delta t)$ of elements $C_{ij} = \langle f_i(t)\, f_j(t + \Delta t)\rangle_t$ of all pairs of features $i$ and $j$ are computed, where $t$ is time, and $\Delta t$ is the tICA lag time. The tICA components (tICs) are linear combinations of features that capture the greatest time-lagged variance, which can be found by maximizing the objective function $\langle \alpha\,|\, C(\Delta t)\,|\, \alpha\rangle$ subject to the constraint that each component has unit variance (i.e. $\langle \alpha_i\,|\, \Sigma\,|\, \alpha_i\rangle = 1$). As features, we computed all pairwise distances between Cα and Cβ atoms (435 and 300 distance pairs for Axin and HIV C-CA binding peptides, respectively) using the MDTraj package[99]. A tICA lag time of $\Delta t = 5$ ns was chosen. After projecting trajectory data to the four largest tICs, conformational clustering was performed using the $k$-centers algorithm to define 50 discrete states for the construction of Markov State Models (MSMs). The GMRQ cross-validation algorithm was used to determine the optimal number of states (Supplementary Fig. 26).

MSM transition matrices $T^{(\tau)}$ were constructed at lag times $\tau$ ranging from 1 to 300 ns, using a maximum-likelihood estimator. Transition matrix elements $T_{ij}^{(\tau)}$ contain the probability of transitioning from state $i$ to state $j$ in time $\tau$. MSM implied timescales are computed as $t_i = \tau/\ln \lambda_i$ where $\lambda_i$ are the eigenvalues of the MSM transition matrix. The implied timescales plateau with increasing lag time, indicating that dyamics is approximately Markovian (i.e. memory-less) beyond a lag time of 100 ns (Supplementary Fig. 27). The slowest implied timescale can be interpreted as the folding/unfolding relaxation time of the peptide, which generally ooccurs along the principal tICA component, $tIC_1$. Example projections of trajectory data to the first two tICs are shown in Supplementary Fig. 28.

**Analysis of secondary structure.** The secondary structure content of simulation snapshots was calculated using the DSSP algorithm implemented in MDTraj[6]. We computed the ensemble average of helicity $\langle h\rangle$ from the trajectory data for each design, using the equilibrium populations $\pi_i$ of each microstate $i$ predicted by the MSM, as $\langle h\rangle = \Sigma_i \pi_i h_i$, where $h_i$ is the average helicity of snapshots belonging to state $i$.

**Statistics and reproducibility.** All the experiments reported here have been repeated independently at least three times, yielding similar results.

**Reporting summary**. Further information on research design is available in the Nature Research Reporting Summary linked to this article.

## Data availability

The data supporting the results and findings of this study are available within the paper and the Supplementary Information files. Additional raw data that support the plots within this paper are available from the corresponding author upon request. All simulated structure files and trajectory data for peptides are available at https://doi.org/10.5281/zenodo.5570769.

## Code availability

All trajectory data and analysis scripts used for simulations have been deposited at https://doi.org/10.5281/zenodo.5570769.

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

## Acknowledgements

We thank the funding from the National Institute of Health (NIH) under the award number R35GM133468 (to R.E.W.), R01GM116204 (to W.Y.), R35GM22552 (to W.Y.), and R01GM123296 (to V.A.V.). We thank the support from the Leukaemia SPORE Career Enhancement Award programme (to R.E.W.). Most peptides were synthesized at the Macromolecular Core Facility at the Penn State University College of Medicine RRID:SCR_020408. We thank Dr. Chris Schafmeister for providing LC–MS access to help our analysis of peptides before and after FTDR coupling. NMR data collection, analysis and discussion with Dr. Charles W. Debrosse, and support for the NMR facility at Temple University by a CURE grant from the Pennsylvania Department of Health are gratefully acknowledged. Circular dichroism (CD) measurements were facilitated by Dr. Allen W. Nicholson. High performance computing resources at Temple University were supported in part by the National Science Foundation through major research instrumentation grant CNS-1625061, the US Army Research Laboratory under contract number W911NF-16-2-0189, and NIH Research Resource computer instrumentation grant S10-OD020095. We also thank the participants of Folding@home, without whom this work would not be possible.

## Author contributions

M.S.I. designed the research, performed the experiments, analysed data, and wrote the manuscript. S.L.J., S.Z., Z.Y.B., Y.G., M.Z. designed the research, performed the experiments, and analysed data. K.H.K., P.K., C.C., R.M. and Z.L. performed the experiments and analysed data. V.A.V., W.Y. and R.E.W. designed and supervised the research and wrote the manuscript.

## Competing interests

The authors declare no competing interests.
