## [Peer Review File · Nature Communications]

REVIEWER COMMENTS

Reviewer #1 (Remarks to the Author):

Very interesting paper by Wang and coworkers. The new stapling technique involving a new reaction based in a fluorine-thiol displacement is relevant and may bring advantages in the peptide field. Although with some limitations (the long stepwise preparation of the modified amino acids, the limited scope presented, the types of bioactivity tested, and that only *i*, *i*+4 cases are analyzed), I believe the manuscript contains valuable scientific findings and may deserve publication after major revisions.

Main Points

-Generalize the scope of the protocol.

a) Determine if cysteine can be present in the sequence, if protection or not of the thiol is needed then. Also it would be interesting to show the compatibility of serine.

b) Although authors state in some cases "unprotected peptides", however I only see C,N-terminal protected derivatives. Why is this happening? Also, to show the power of the protocol, stapling of cyclopeptides may be checked or, eventually, the N-C-macrocyclization of the stapled linear peptides may be explored. This would add value to the paper.

c) State in the main text the base used to generate the thioate nucleophile (it only appears in the Methods section). As it is NaOH, I find authors should determine or justify with strong arguments that epimerization does not take place under the conditions used for stapling. In this respect, in the Supplemental Information, the HPLC profiles of the stapled peptides are a little broad. On the other hand the experiments with the combinations of LD peptides may be analyzed or commented in this context to shed some light.

- Complement the structural assignment (CD, computational techniques) with NMR studies. This is very important, as all assumptions on the helicity will have an atomistic back up. Also comparisons with the RCM peptides will be clarified. A complete description of relevant stapled peptides with high field NMR experiments (bidimensional, NOEs, temperature dependent shifts, etc.) is needed and it would be very illustrative.

- I find that the introduced technique is very valuable and can be generally used for stapling, but I don't find enough conclusive arguments to say it is better than alternative protocols. In my opinion, to say this, authors should make more experimentation with other sequences, and thoroughly check different bioactivities and properties. The present paper only shows limited examples with, in some cases, minor differences between the control RCM peptides and the new stapled ones. The text should reflect this point. See for instance, in the Abstract "the cellular uptake ...was universally enhanced....". Authors should reconsider the whole text and modify it accordingly, no need to pretend surpass the RCM stapled peptides with a limited set of determinations. The fluorine displacement reaction should be considered another (valuable, promising, general if demonstrated) method.

Minor points

-Please, cite some references where the nucleophilic displacement from a thiolate is used in stapling: Harran octafluoropentene (Angew. Chem. Int. Ed. 2020, 59, 674 –678) and the Wilson double click (Chem. Sci., 2017, 8, 5166) methods. Perhaps authors may comment the complementary strategy of their method in comparison with the well established alkylation of cysteines for stapling.

-Authors talk in several parts of the text about "more nucleophilic thiolates". In particular citing refs 36, 37 and 41. I don't see why benzyl thiol should be more nucleophilic than others. In my opinion, bulky derivatives or conjugated/aromatic thiols part, almost all alkyl thiols should be similar nucleophiles. In fact, I could not find these differences in the mentioned references. Please, check and eventually, modify the text.

-Authors may consider alternative stapling modes (*i*, *i*+7; others?). Comment and/or explore.

In summary, an excellent paper, which contains valuable scientific discoveries, with clear usefulness for the community, but needing revisions for publication. The manuscript is well written, although I find it a little overemphasized in some aspects (without the necessary experimental support). However, if additional determinations are done, and the manuscript is rewritten, I consider it may be publishable in Nat. Comm.

Reviewer #2 (Remarks to the Author):

Wang and colleagues report a macrocyclization reaction for unprotected peptides employing fluoroacetamide containing amino acids as electrophiles and bis-thiols as the nucleophile. The cyclization reaction is initially optimized using a short test sequence and then applied to three more peptides (two of them related) that have been previously used for stapling approaches. The authors show that the "Fluorine-Thiol Displacement Reaction (FTDR)" (if the appropriate amino acids/crosslinker combination is used) can increase helicity and cellular uptake of peptides. In addition the authors, validate their observations regarding the stabilization effect via MD simulations and investigate the uptake mechanisms in detail. The latter is a notable effort that is only rarely performed for such peptides but important to evaluate their usability and support future optimization campaigns.

The study addresses a central limitation of peptide derived inhibitors, their often low cellular uptake, and introduces a novel crosslinking reaction. This in principle renders the manuscript interesting and potentially suitable for publication in Nat. Commun. However, to really contribute to the field it is necessary to i) show the generality of the approach (broad sequence scope and general effect on cellular uptake) and/or ii) validate the development of an inhibitor with a clearly improved bioactivity (see details below). For that reason, the following points should be addressed by the authors:

Major:

1) The authors claim that FTDR-based coupling represents an efficient macrocyclization approach in synthesizing cyclic unprotected peptides with flexible linker choices. Considering, that only four peptide sequences with only a limited set of amino acids were tested this statement is not fully supported. In particular, the limitation associated with the presence of cysteine have not been addressed. For that reason, the presence of cysteine in the peptide with varying distance to the XL/XD site should be investigated.

2a) To support the beneficial effect of this new staple a more systematic investigation of uptake behaviour should be performed.

and/or

2b) The improved activity on beta-catenin and the Wnt signalling pathway should be confirmed by a reporter gene assay or target gene analysis, since effects on the viability of DLD-1 cells may be the result of off target effects.

Minor:

3) Significant numbers for yields should be reduced to two e.g. in Table S1 and in the text (so 53% instead of 52.7%)

4) When providing yields it should be clarified if this is for macrocyclization or total yield.

5) Manuscript Table 1 should also state yields

6) the use of "enantiomer" and "chirality" should be revised as once the chiral building blocks are incorporated in the peptides, inversion of stereo center(s) results in epimer/diastereomer.

7) For the modification with different XL/XD combinations the authors write "...seemed to bring in more rigid conformations..." Changes in rigidity are one possible explanation (among others). Should be removed as to speculative.

8) The authors write: "Notably, a direct comparison of the fluorescence signals for all the peptides at 50 μ M or 200 μ M doses seemed to pinpoint the potentially better binding of FTDR stapled peptides than the RCM control 51." It is not clear why this would be the case, as the EC50 should be the value to evaluate affinity (which the authors do in the previous sentence). Total intensity can differ for various reasons.

9) The authors write "To our best knowledge, this is also the first reported effort to elucidate the cellular uptake mechanism of peptides stapled by strategies other than RCM." In this respect, please see J. Am. Chem. Soc. 2020, 142, 14461–14471 and references therein.

Reviewer #3 (Remarks to the Author):

Wang, et.al. reported a fluorine-thiol displacement reaction which was demonstrated for unprotected peptide macrocyclization and stapling. The authors screened different thiol linkers and found 1, 3- benzenedimethanethiol linker as the optimal linker for the macrocyclization and stapling strategy based on a fluorine thiol displacement reaction. This reaction provides good chemoselectivity and functional group tolerance in mild temperature, but the product yields are not good enough. The stapling strategy also was used in stapling peptides at i, i+4 positions promoting alpha helicity of a variety of peptide substrates, which is comparable to the control peptides, classic ring-closing metathesis (RCM) stapled peptides.

One feature of these stapled peptides is that the fluorine-thiol displacement derived peptides showing improved cell permeability and good cellular uptake. The authors also elaborated in pilot mechanism studies of the cellular uptake, suggesting the uptake of FTDR-stapled peptides may

involve multiple endocytosis pathways. The improved cellular uptake renders the stapled Axin peptide with good stability, affinity, and inhibition of cancer cell growth. In general, the work is of good quality and should be interesting to the peptide stapling and drug discovery fields, but the following issues still need to be addressed:

1. The method of peptide synthesis should be provided more detailed. Are modified amino acids 15 or 16 compatible in traditional fmoc-based solid-phase chemistry procedures? Or still need further derivation after the peptide synthesis process?
2. The cytotoxicity of the developed peptides was demonstrated with DLD-1 cells which as used for growth inhibition later in the manuscript. However, the cells were only incubated in 12 hours, the period of the incubation is too short, these results is not convincing. One of the control experiment is that the authors should display the cytotoxicity of the developed peptides in different time points with longer incubation, and also should measure the cell cytotoxicity in other kind of cells such as 293T cells or other cell without the minimal effect from the binding of the peptide with target protein b-catenin in longer incubation time, probably up to 2-3 days, and then measure the cell viability.
3. The halogen-thiol displacement reactions were reported in stapling strategies previous in literature (Angew. Chem. Int. Ed. 2018, 57, 11164-11170, and Chem. Commun., 2005, 2552-2554). Is there any notable advantage of the fluorine-thiol displacement reaction with modified unnatural amino acids (compounds 15 and 16), compared the above mentioned stapling strategies, where the cysteines incorporated in targeted peptides react with chloroacetamide or bromoacetamide derived linkers. Because those peptides are really unprotected peptides, while the peptides used in current manuscript were modified with unnatural amino acids (15 or 16). The authors should discuss the comparison of those strategies or the potential advantage of the fluorine-thiol displacement reaction, which would be of interest to others in the community.
4. Because of the relative low reactivity of the fluoroacetamide, the stapling reactions were carried out in basic buffer and the reaction time is up to 12 hours, the reaction rates and reaction conditions are not perfect, compared to other clickable stapling strategies. The two cysteines in targeted peptides react with N,N'-(1,4-phenylenebis(methylene))bis(2-chloroacetamide) or N,N'-(1,4-phenylenebis(methylene))bis(2-bromoacetamide) may provide with good reaction rates and under even milder conditions, providing the structural similarity. Comparison of this reaction to FTDR method with respect to their efficiency in peptide stapling is necessary.
5. In the method section of "General procedures for peptide stapling", the pH value of the reactions should be provided exactly, does the pH value more basic than pH=9 in those reaction mixtures?
6. In Supplementary Information, the LC trace, MS-Spectra of all the linear peptides are missing. The original LC trace, MS-Spectra of the linear peptides should be provided in supplementary data.

October 21, 2021

Enclosed please find the **revised** manuscript entitled "Unprotected Peptide Macrocyclization and Stapling via A Fluorine-Thiol Displacement Reaction", for consideration towards publication in *Nature Communications* as a research article. We would like to **express our gratitude** to all three reviewers for their insightful comments and suggestions.

Major revisions have been made with incorporation of new experimental data (Main Text: Figure 2, Figure 4, Figure 7h, Figure 8, Table 1; Supplementary Information: Tables S1 – S5, Figures S4 – S25, Figures S36 - S37, Figures S41 - S42 and updated LC-MS spectra raw data). A copy of the manuscript and the supplementary information with annotations (highlighted in **yellow**) of the major revisions have been uploaded and submitted. Please find our point-by-point responses (**in blue**) to reviewers' comments/questions below:

Reviewer(s)' Comments to Author:

Reviewer 1

Very interesting paper by Wang and coworkers. The new stapling technique involving a new reaction based in a fluorine-thiol displacement is relevant and may bring advantages in the peptide field. Although with some limitations (the long stepwise preparation of the modified amino acids, the limited scope presented, the types of bioactivity tested, and that only i, i+4 cases are analyzed), I believe the manuscript contains valuable scientific findings and may deserve publication after major revisions.

Main Points

-Generalize the scope of the protocol.

a) Determine if cysteine can be present in the sequence, if protection or not of the thiol is needed then. Also it would be interesting to show the compatibility of serine.

We would like to thank the reviewer for suggesting this important experiment. Our previous small-molecule model reactions with benzyl thiol linkers appeared to proceed faster and more efficiently than the same model substrate's reaction with cysteine, suggesting that the benzenedimethanethiol-mediated FTDR could compete against cysteine. During the past 12 months of revisions, we have attempted FTDR on another model peptide that has serine but also cysteine at different positions (**20 – 22, 26**, Figure 2). Using LC-MS, we observed that FTDR-cyclization proceeded smoothly and generated the desired major product, tolerating the presence of unprotected cysteine (Figure S4). We also attempted N,C-terminal FTDR cyclization on a 14mer long sequence (**28, 30, 32**) which has cysteine(s) present at different sites and have successfully isolated the products. We suspect that our two-step FTDR approach (deprotonation of the benzenedimethanethiol

linker before adding it to the peptide mixture) also helped promote the crosslinker-mediated FTDR. We have thereby added Figure 2, Figure S4, updated Table 1 and Table S1. We also added the phrase “in the presence of intrinsic cysteines” to the Abstract, the phrase “sparing intrinsic cysteines” to the Discussion Section, and the related background into the Introduction Section (2nd paragraph, 3rd paragraph). The Results Section has been accordingly updated at the beginning of the 2nd paragraph (“To explore the compatibility with cysteines...”) and the 3rd paragraph (“Structurally, a 14-mer rationally...”).

b) Although authors state in some cases “unprotected peptides”, however I only see C,N-terminal protected derivatives. Why is this happening? Also, to show the power of the protocol, stapling of cyclopeptides may be checked or, eventually, the N-C-macrocyclization of the stapled linear peptides may be explored. This would add value to the paper.

We thank the reviewer for catching this – we usually put N-terminal Ac and C-terminal Amide as a default setting for peptide synthesis. To show that FTDR stapling is flexible with protected or deprotected N,C-terminals, we now have synthesized model peptides **20-22** with both terminals unprotected (Figure 2a) as were the newly synthesized model Axin peptides for NMR studies (**57-59**, Figure 4a). Other recently-synthesized new peptides such as **26**, **28**, **30**, and **32** also have free N-terminal but C-terminal amide as the core synthesis facility only has the carboxylate-generating resins that have been pre-coupled with natural amino acids. Generally speaking, FTDR-based macrocyclization worked well with all these peptides as evidence by Figure S4, LC-MS of the final desired products (available at the end of the Supplementary Information) and Table 1/Table S1.

Following the reviewer’s suggestion, we indeed found that DeGrado et al. have published a N-C-macrocyclization to staple a 14-mer long linear peptide (PMID: 29417711), and thereby adopted this sequence (with insertion of cysteines to further demonstrate our approach’s tolerance of cysteines) for our FTDR-mediated terminal cyclization. The new experiments led to a class of stapled peptides as evidenced by the new supplementary Figure S5. Accordingly, we also updated Table 1/Table S1, and added results to the third paragraph in the Results Section, starting with “Structurally, a 14-mer rationally designed peptide ...”.

c) State in the main text the base used to generate the thioate nucleophile (it only appears in the Methods section). As it is NaOH, I find authors should determine or justify with strong arguments that epimerization does not take place under the conditions used for stapling. In this respect, in the Supplemental Information, the HPLC profiles of the stapled peptides are a little broad. On the other hand the experiments with the combinations of LD peptides may be analyzed or commented in this context to shed some light.

We are sorry for the confusion, and totally agree that more details are needed to better illustrate our FTDR experiments on peptides. The thiolates were actually first incubated

with a less than stoichiometric amount of NaOH to ensure their deprotonation but also to avoid creating an overly basic environment. The deprotonated crosslinker was then added to the peptide solution as the 2nd step. We have now tested the final reaction mixtures' pH and found that most have a pH range of 9.0 – 9.5. Within this pH range and under the mild reaction temperature, amino acid epimerization within the peptides should not happen at a significant level (PMID 8058648, PMID 7537425). We appreciate the reviewer pointing out some of our stapled peptides' broad HPLC profiles, which we figured out was actually due to a leak in the old LC-MS instrument. To overcome this non-scientific issue, we repeated the LC-MS analysis and were able to obtain sharp LC peaks.

Thus, we now added “a final pH of 9.0 - 9.5” to the Methods Section and provided more detailed explanations about our FTDR on peptides including the base NaOH in the first paragraph of the Results Section. We also justified racemization in the fourth paragraph of the Results Section with “The reaction mixtures had final pH of ...”. Experimentally, on top of replacing LC spectra in Supplementary Information, we also did thorough NMR studies of the representative stapled peptides that only differed in the L/D chiral substrates and showed distinct NMR patterns. We also attempted chiral separation of these peptides and showed that they each possess a different retention time in HPLC. These experiment results are available now as Figure 4, Tables S2-S4, Figures S6-S23, and importantly, Figure S25. Discussion of these experimental results has been added as the 2nd paragraph under the subtitle “Linker and Chirality requirement in FTDR-based i,i+4 stapling”.

- Complement the structural assignment (CD, computational techniques) with NMR studies. This is very important, as all assumptions on the helicity will have an atomistic back up. Also comparisons with the RCM peptides will be clarified. A complete description of relevant stapled peptides with high field NMR experiments (bidimensional, NOes, temperature dependent shifts, etc.) is needed and it would be very illustrative.

As mentioned above, we now have performed NMR studies of the three representative Axin peptides. Results can be found as Figure 4, Table 1, Table S1-S4, Figure S6-S24. We also dedicated one new paragraph to discuss these NMR results (2nd paragraph under the subtitle “Linker and Chirality requirement in FTDR-based i,i+4 stapling”). Indeed, the NMR analysis complemented other experimental data such as CD and simulation, confirming that FTDR-stapling at X_L,X_L; or X_L,X_D substrates promoted peptide folding much more significantly than it did to X_D,X_L substrates.

- I find that the introduced technique is very valuable and can be generally used for stapling, but I don't find enough conclusive arguments to say it is better than alternative protocols. In my opinion, to say this, authors should make more experimentation with other sequences, and thoroughly check different bioactivities and properties. The present paper only shows limited examples with, in some cases, minor differences between the control RCM peptides and the new stapled ones. The text should reflect this point. See for instance, in the Abstract “the cellular uptake ...was universally enhanced...”. Authors should reconsider the whole text and modify it accordingly, no need to pretend surpass the RCM stapled peptides with a limited set of determinations. The fluorine displacement reaction should be considered

another (valuable, promising, general if demonstrated) method.

We appreciate the reviewer's encouragement and suggestions. Over the past 12 months, we have thereby attempted FTDR stapling on two more classes of peptide sequences (SAHB_A, STAD) for i,i+4 linkage and on the classical p53 peptide sequence for i,i+7 linkage. With more experimental studies on their stapling effects, cellular uptakes of the resulting peptides, along with in vitro and in vivo efficacies (for Axin and p53 peptides), we feel the fluorine displacement reaction (FTDR) may be considered a promising general method to promote peptide folding and activities. Results for SAHB_A and STAD peptides have now been added as Figure S36 and Figure S37, respectively and were discussed in the third paragraph under the subtitle "Cell permeability of i,i+4 FTDR-stapled peptide analogues" in the main text. Additional biological characterizations of lead Axin peptides were added as Figure 7h and Figure S41, S42a and were discussed in the paragraph titled "Activity of i,i+4 FTDR-stapled peptide analogues". The biological characterizations of p53 peptides have been presented in Figure 8 and Figure S42b. Discussion of their activities including the enhanced uptake can be found in the last two paragraphs in the Results Section that are subtitled as "Exploration of FTDR stapling at i,i+7 positions".

Minor points

-Please, cite some references where the nucleophilic displacement from a thiolate is used in stapling: Harran octafluoropentene (*Angew. Chem. Int. Ed.* 2020, 59, 674–678) and the Wilson double click (*Chem. Sci.*, 2017, 8, 5166) methods. Perhaps authors may comment the complementary strategy of their method in comparison with the well established alkylation of cysteines for stapling.

Cited and commented on the 2nd paragraph of the Introduction Section.

-Authors talk in several parts of the text about "more nucleophilic thiolates". In particular citing refs 36, 37 and 41. I don't see why benzyl thiol should be more nucleophilic than others. In my opinion, bulky derivatives or conjugated/aromatic thiols part, almost all alkyl thiols should be similar nucleophiles. In fact, I could not find these differences in the mentioned references. Please, check and eventually, modify the text.

We agree that although these refs highlighted great activity of the benzyl thiol, there is no real evidence that it is more nucleophilic than other alkyl thiols. As mentioned above, our stepwise peptide modification strategy that pre-deprotonated benzyl thiols at the first step may help with the reactivity. More importantly, the reported F- π interactions (PMID 27145463, PMID 28464551) between the fluorine in the substrate and the benzene ring of the benzyl thiol may bring the thiol to closer proximity and thereby facilitate the fluorine displacement reaction. We have revised the wording in the main text and cited these literature (The first paragraph in the Results Section and the 2nd paragraph under the subtitle "Substrate scope with various linkers").

-Authors may consider alternative stapling modes (i, i+7; others?). Comment and/or explore.

Following the reviewer's suggestion, we have hereby attempted stapling at i,i+7 on p53 peptides. Related data (as detailed above) have been incorporated into the manuscript and discussed in the Results Section. Both the Abstract and the Discussion Section have been also updated accordingly. Briefly, we believe that the FTDR stapling on i,i+7 further demonstrated its effect on stapling diverse sequences and endowing peptides with enhanced folding and cellular activities.

In summary, an excellent paper, which contains valuable scientific discoveries, with clear usefulness for the community, but needing revisions for publication. The manuscript is well written, although I find it a little overemphasized in some aspects (without the necessary experimental support). However, if additional determinations are done, and the manuscript is rewritten, I consider it may be publishable in Nat. Comm.

We appreciate all the encouragements and have added a good amount of new experiments according to the suggestions, which hopefully can better support the discussions and conclusions in the manuscript that have been also rewritten.

Reviewer 2

Wang and colleagues report a macrocyclization reaction for unprotected peptides employing fluoroacetamide containing amino acids as electrophiles and bis-thiols as the nucleophile. The cyclization reaction is initially optimized using a short test sequence and then applied to three more peptides (two of them related) that have been previously used for stapling approaches. The authors show that the "Fluorine-Thiol Displacement Reaction (FTDR)" (if the appropriate amino acids/crosslinker combination is used) can increase helicity and cellular uptake of peptides. In addition the authors, validate their observations regarding the stabilization effect via MD simulations and investigate the uptake mechanisms in detail. The latter is a notable effort that is only rarely performed for such peptides but important to evaluate their usability and support future optimization campaigns.

The study addresses a central limitation of peptide derived inhibitors, their often low cellular uptake, and introduces a novel crosslinking reaction. This in principle renders the manuscript interesting and potentially suitable for publication in Nat. Commun. However, to really contribute to the field it is necessary to i) show the generality of the approach (broad sequence scope and general effect on cellular uptake) and/or ii) validate the development of an inhibitor with a clearly improved bioactivity (see details below). For that reason, the following points should be addressed by the authors:

We thank the reviewer for all the encouragements and have carefully followed the points suggested below to revise the manuscript.

Major:

1) The authors claim that FTDR-based coupling represents an efficient macrocyclization approach in synthesizing cyclic unprotected peptides with flexible linker choices. Considering, that only four peptide sequences with only a limited set of amino acids were tested this statement is not fully supported. In particular, the limitation associated with the presence of cysteine have not been addressed. For that reason, the presence of cysteine in the peptide with varying distance to the XL/XD site should be investigated.

We appreciate these insightful comments and have thereby attempted FTDR stapling on a different model peptide sequence with cysteines at varying distances to the XL/XL sites (peptide **20 – 22, 26**, Figure 2). Further, we performed N,C-terminal FTDR cyclization on a 14mer long sequence (**28, 30, 32**) which has cysteine(s) present at varying distances to the unnatural crosslinker site. LC-MS based reaction monitoring demonstrated successful conversion by FTDR (Figure S4) resulting in the desired products. We believe that the benzyl thiol-based crosslinkers dominated the FTDR and competed off the cysteines in presence due to our stepwise pre-deprotonation and the reported F- π interactions (PMID 27145463, PMID 28464551). We now have updated the results with well characterized macrocyclization products in Table 1, Table S1, and the raw LC-MS spectra at the end of the Supplementary Information. In the main text, the phrase “in the presence of intrinsic cysteines” has been added to the Abstract, and the phrase “sparing intrinsic cysteines” has been incorporated to the Discussion Section. The Introduction Section (2nd paragraph, 3rd paragraph) and the Results Section (the 1st paragraph, the beginning of the 2nd paragraph, the 3rd paragraph, the very end of the 4th paragraph) have been also updated accordingly. For functional peptide studies, we now have added two more classes of peptide sequences (SAHB_A, STAD) for i,i+4 linkage and the classical p53 peptide sequences for i,i+7 linkage. Results have been discussed in the main text (the third paragraph under the subtitle “Cell permeability of i,i+4 FTDR-stapled peptide analogues”, and the last two paragraphs under the subtitle “Exploration of FTDR stapling at i,i+7 positions”). In summary, FTDR coupling on five more types of peptide sequences have been further demonstrated, and the cysteine presence seems to be tolerated.

2a) To support the beneficial effect of this new staple a more systematic investigation of uptake behavior should be performed.

We completely agree with the reviewer and have since studied FTDR stapling on three more types of functional peptides over the past 12 months. The cellular uptake results of stapled SAHB_A analogues and stapled STAD peptide mimics have been added as Figure S36 and Figure S37 in the Supplementary Information and were discussed in the newly-added third paragraph under the subtitle “Cell permeability of i,i+4 FTDR-stapled peptide analogues” in the main text. The uptake results of the stapled p53 peptide analogues have been added into Figure 8. The related discussions can be found as the last two paragraphs in the Results Section that are subtitled as “Exploration of FTDR stapling at i,i+7 positions”. Based on these results, we think the new staple could afford an enhanced cell uptake for multiple different types of peptide sequences.

and/or 2b) The improved activity on beta-catenin and the Wnt signalling pathway should be confirmed by a reporter gene assay or target gene analysis, since effects on the viability of DLD-1 cells may be the result of off target effects.

We would like to thank the reviewer for pointing this out. The TOPFlash luciferase reporter assay in DLD-1 cells has been thereby performed and we have observed the down-regulation by FTDR-stapled Axin derivatives on β -catenin mediated transcription (Figure S42a), which is consistent with the previously observed viability assay results. Inspired by the reviewer's comments, we also performed the reporter assay for the newly developed p53 stapled peptides (Figure S42b), and showed that our stapled peptides can upregulate the p53-responsive transcription. The related discussions can be found in the paragraph titled "Activity of i,i+4 FTDR-stapled peptide analogues" and also in the last paragraph of the Results Section that described the activities of the i,i+7 stapled p53 peptides.

Minor:

3) Significant numbers for yields should be reduced to two e.g. in Table S1 and in the text (so 53% instead of 52.7%)

Corrected as suggested.

4) When providing yields it should be clarified if this is for macrocyclization or total yield.

We have now changed the "yield" to "cyclization yield" at the end of the 1st paragraph (Results Section) and the middle of the 2nd paragraph (Results Section). We further changed the wording and added "macrocyclization yields" to the 3rd paragraph under the subtitle "Cell permeability of i,i+4 FTDR-stapled peptide analogues", the 2nd paragraph under the subtitle "General Procedures for Peptide Stapling", and Table 1 title, as well as Table S1 title.

5) Manuscript Table 1 should also state yields

Added as suggested.

6) The use of "enantiomer" and "chirality" should be revised as once the chiral building blocks are incorporated in the peptides, inversion of stereo center(s) results in epimer/diastereomer.

We believe that most of the chiral building blocks after incorporation into peptides should still retain the original enantiomeric chirality due to the coupling method we adopted (HATU, DIPEA), which according to the literature (*J. Am. Chem. Soc.* 1993, 115, 4397-4398) has lowered the extent of racemization to less than 1-2%. We are sorry for causing these confusions, and have reworded the sentence about Peptide Synthesis in the Methods Section, adding the phrase "including compound 15 for X_L or compound 16 for X_D", clarifying that "has been reported to minimize racemization" with the citation of this reference.

7) For the modification with different XL/XD combinations the authors write "...seemed to bring in more rigid conformations..." Changes in rigidity are one possible explanation (among others). Should be removed as to speculative.

Thanks for catching this and we have removed it as suggested.

8) The authors write: "Notably, a direct comparison of the fluorescence signals for all the peptides at 50 μ M or 200 μ M doses seemed to pinpoint the potentially better binding of FTDR stapled peptides than the RCM control 51." It is not clear why this would be the case, as the EC50 should be the value to evaluate affinity (which the authors do in the previous sentence). Total intensity can differ for various reasons.

Agreed, and we have now removed this sentence.

9) The authors write "To our best knowledge, this is also the first reported effort to elucidate the cellular uptake mechanism of peptides stapled by strategies other than RCM." In this respect, please see J. Am. Chem. Soc. 2020, 142, 14461–14471 and references therein.

Although there have been reported mechanism studies for crosslinked and macrocyclized peptides, there was no uptake mechanism study for stapled peptides (stabilized α -helix) until this JACS paper (PMID 32786217) came out. We thank the reviewer for catching this and have accordingly changed our claim in the Discussion Section (the 2nd paragraph) to "there have been few reported efforts...", and have acknowledged and cited this JACS reference with the new sentence "During the manuscript review and revisions, several peptidomimetics have been...".

Reviewer 3

Wang, et.al. reported a fluorine-thiol displacement reaction which was demonstrated for unprotected peptide macrocyclization and stapling. The authors screened different thiol linkers and found 1, 3- benzenedimethanethiol linker as the optimal linker for the macrocyclization and stapling strategy based on a fluorine thiol displacement reaction. This reaction provides good chemoselectivity and functional group tolerance in mild temperature, but the product yields are not good enough. The stapling strategy also was used in stapling peptides at i, i+4 positions promoting alpha helicity of a variety of peptide substrates, which is comparable to the control peptides, classic ring-closing metathesis (RCM) stapled peptides.

One feature of these stapled peptides is that the fluorine-thiol displacement derived peptides showing improved cell permeability and good cellular uptake. The authors also elaborated in pilot mechanism studies of the cellular uptake, suggesting the uptake of FTDR-stapled peptides may involve multiple endocytosis pathways. The improved cellular

uptake renders the stapled Axin peptide with good stability, affinity, and inhibition of cancer cell growth.

In general, the work is of good quality and should be interesting to the peptide stapling and drug discovery fields, but the following issues still need to be addressed:

1. The method of peptide synthesis should be provided more detailed. Are modified amino acids 15 or 16 compatible in traditional fmoc-based solid-phase chemistry procedures? Or still need further derivation after the peptide synthesis process?

Yes, the modified amino acid building blocks 15 or 16 were compatible with traditional solid phase synthesis and were directly incorporated into the peptides following the HATU/DIPEA – Piperidine mediated Fmoc-based chemistry coupling procedures. No further derivatization is needed. We are sorry for any confusions caused by the previous version of manuscript and have since revised the “Peptide Synthesis” description in the Methods Section with more details, e.g. with the addition of the phrase “including compound 15 for X_L or compound 16 for X_D” and the clarification of racemization with the citation of literature (*J. Am. Chem. Soc.* 1993, 115, 4397-4398) for the use of HATU/DIPEA. We added the statement that “15 for X_L, 16 for X_D, see Supplementary Information for detailed synthesis” and “by Fmoc-based solid-phase synthesis” into the Results Section, the 2nd paragraph under the subtitle “Substrate scope with various linkers”. More detailed descriptions of the follow-up FTDR crosslinking on peptides were also added within the same paragraph, and also into the 1st paragraph of the Results Section.

2. The cytotoxicity of the developed peptides was demonstrated with DLD-1 cells which as used for growth inhibition later in the manuscript. However, the cells were only incubated in 12 hours, the period of the incubation is too short, these results is not convincing. One of the control experiment is that the authors should display the cytotoxicity of the developed peptides in different time points with longer incubation, and also should measure the cell cytotoxicity in other kind of cells such as 293T cells or other cell without the minimal effect from the binding of the peptide with target protein b-catenin in longer incubation time, probably up to 2-3 days, and then measure the cell viability.

Following the reviewer’s suggestions, we now have performed these control experiments by titrating the cytotoxicity of the Axin peptides at different time points (24h, 48h, and 96h) (Figure S41). Consistent with the results at 12h, no significant effects on viability were observed for 24h incubation. Nevertheless, the inhibition effects by FTDR-stapled peptides **67** and **65** were significant after 48h or 96h incubation, which was consistent with the results observed after 120h incubation. Taken together, these results suggest that FTDR-stapled Axin leads inhibited more of the cell growth. We also performed the growth inhibition assay using the same set of peptides on HEK293T cells which have reported low basal level of beta-catenin (Figure 7h). As expected, little growth inhibition was observed, suggesting that the growth inhibition effects of these peptides depended on inhibition of beta-catenin mediated Wnt signaling. We have discussed these results in the Results Section, at the end of the paragraph that is subtitled as “Activity of i,i+4 FTDR-stapled

3. The halogen-thiol displacement reactions were reported in stapling strategies previous in literature (Angew. Chem. Int. Ed. 2018, 57, 11164-11170, and Chem. Commun., 2005, 2552-2554). Is there any notable advantage of the fluorine-thiol displacement reaction with modified unnatural amino acids (compounds 15 and 16), compared the above mentioned stapling strategies, where the cysteines incorporated in targeted peptides react with chloroacetamide or bromoacetamide derived linkers. Because those peptides are really unprotected peptides, while the peptides used in current manuscript were modified with unnatural amino acids (15 or 16). The authors should discuss the comparison of those strategies or the potential advantage of the fluorine-thiol displacement reaction, which would be of interest to others in the community.

We would like to thank the reviewer for this insightful suggestion. Direct crosslinking with bromo/chloro-alkyl chemistry has been known to result in over-alkylation with multiple equivalents of the halogen crosslinker into the same peptide, particularly at the same or different nucleophilic amino side chains causing cross-reactivity (Chem.Sci., 2014.5, 1804-1809). Moreover, modification of the unprotected cysteines will prohibit the application of their halogen-thiol displacement strategies to stapling peptides that actually need free cysteines to participate important protein-protein interactions or to retain the desired structures/functions (PMID 23362256, PMID 9669552, PMID 28326775). During this revision, we have systematically explored our unnatural amino acid-based fluorine-thiol displacement reaction on multiple model peptide sequences (Figure 2) and demonstrated that just by using the fluorine containing unnatural amino acids in the target peptides, we can perform selective crosslinking of our linker with fluorine even in the presence of cysteines. Thus, we added these new results with Figure 2, Figure S4, updated Table 1, and Table S1. We also added the phrase “in the presence of intrinsic cysteines” to the Abstract, the phrase “sparing intrinsic cysteines” to the Discussion Section, and the related background discussion into the Introduction Section (2nd paragraph with the citation of the two references the reviewer mentioned, 3rd paragraph). The Results Section has been accordingly updated at the beginning of the 2nd paragraph (“To explore the compatibility with cysteines...”) and the 3rd paragraph (“Structurally, a 14-mer rationally...”).

4. Because of the relative low reactivity of the fluoroacetamide, the stapling reactions were carried out in basic buffer and the reaction time is up to 12 hours, the reaction rates and reaction conditions are not perfect, compared to other clickable stapling strategies. The two cysteines in targeted peptides react with N,N'-(1,4-phenylenebis(methylene))bis(2-chloroacetamide) or N,N'-(1,4-phenylenebis(methylene))bis(2-bromoacetamide) may provide with good reaction rates and under even milder conditions, providing the structural similarity. Comparison of this reaction to FTDR method with respect to their efficiency in peptide stapling is necessary.

We agree with the reviewer that the current FTDR reaction has room to be further optimized in terms of reaction pH and the reaction time. Nevertheless, the FTDR reaction appears to be more chemoselective and can allow for the presence of unprotected cysteines.

As mentioned above (in response to the question 3), we have carried out major revisions in both experiments and also writing to compare the FTDR method with the reported cysteine – chloro/bromo coupling approach. We now have also discussed the potential of improving FTDR reaction conditions in the Discussion Section, the end of the 2nd paragraph.

5. In the method section of “General procedures for peptide stapling”, the pH value of the reactions should be provided exactly, does the pH value more basic than pH=9 in those reaction mixtures?

Corrected as suggested. We measured the pH values for all those reaction mixtures and they are in the range of pH 9.0-9.5. This detail has been added to the Method Section, but also the Results Section, and the Legends for Figure 1 and Figure 2.

6. In Supplementary Information, the LC trace, MS-Spectra of all the linear peptides are missing. The original LC trace, MS-Spectra of the linear peptides should be provided in supplementary data.

Thanks for pointing this out. We have now added the LC traces and the MS spectra for all the linear peptides into the Supplementary Information (at the end).

In conclusion, we greatly appreciate all the reviewers’ comments and suggestions, which we believe have helped to further improve this research article and make it suitable for publication.

Sincerely,

Rongsheng (Ross) E. Wang

Rongsheng(Ross) Wang, Ph.D.
Assistant Professor
Department of Chemistry
Temple University

Email: rosswang@temple.edu
Tel: (215) 204-1855
Cell: (314) 341-0544

REVIEWERS' COMMENTS

Reviewer #1 (Remarks to the Author):

After the new research done, which addresses all requests from the referees, I believe the manuscript is ready for publication.

In summary, the scope of the process has been substantially increased (along the lines that were considered meaningful and relevant), the physical characterization (through NMR techniques) of the stapled peptides has also been suitably determined and the biological aspects seem, in my opinion, well resolved.

The minor points have also been looked at.

I think that with the revision, it is a very good manuscript, describing a new stapling methodology with a broad application in peptide chemistry that may attract the interest of the community and has potential to be applied in several contexts.

Reviewer #2 (Remarks to the Author):

In their revised version of the manuscript, the authors have addressed all reviewer comments adequately. I can recommend publication of this manuscript.

Please note the following minor point: When referring to the inversion of a stereo-center in a peptide, it should be called "epimerization" not "racemization" (it's not a racemate that is formed but a mixture of different diastereomers)!

Reviewer #3 (Remarks to the Author):

The authors have addressed most of my comments. The revised manuscript was improved in quality. And I would recommend it for publication in Nat. Commun.

November 25, 2021

Enclosed please find the **revised** manuscript entitled "Unprotected Peptide Macrocyclization and Stapling via A Fluorine-Thiol Displacement Reaction", for consideration towards publication in *Nature Communications* as a research article. We would like to **express our gratitude** to all the reviewers for their support

Minor revisions have been made to replace the word "racemization" with "epimerization" throughout the manuscript. A copy of the manuscript and the supplementary information with annotations (highlighted in **yellow**) of the major revisions have been uploaded and submitted, along with the manuscript and SI that have all the revisions accepted. Please find our point-by-point responses (**in blue**) to reviewers' comments/questions below:

Reviewer(s)' Comments to Author:

Reviewer 1

After the new research done, which addresses all requests from the referees, I believe the manuscript is ready for publication.

In summary, the scope of the process has been substantially increased (along the lines that were considered meaningful and relevant), the physical characterization (through NMR techniques) of the stapled peptides has also been suitably determined and the biological aspects seem, in my opinion, well resolved.

The minor points have also been looked at.

I think that with the revision, it is a very good manuscript, describing a new stapling methodology with a broad application in peptide chemistry that may attract the interest of the community and has potential to be applied in several contexts.

We would like to thank the reviewer for the supportive comments and all previous suggestions.

Reviewer 2

In their revised version of the manuscript, the authors have addressed all reviewer comments adequately. I can recommend publication of this manuscript.

Please note the following minor point: When referring to the inversion of a stereo-center in a peptide, it should be called "epimerization" not "racemization" (it's not a racemate that is formed but a mixture of different diastereomers)!

Thanks for catching this. We now have revised the main text and replaced the word “racemization” with “epimerization” on page 6, page 9, and page 17. We thank the reviewer for recommending the publication of this manuscript.

Reviewer 3

The authors have addressed most of my comments. The revised manuscript was improved in quality. And I would recommend it for publication in Nat. Commun.

We appreciate the reviewer’s recommendation and wanted to thank the reviewer for all previous comments and suggestions.

In conclusion, we are grateful to all the reviewers for their comments and suggestions, which we believe have helped to further improve this research article and make it suitable for publication.

Sincerely,

Rongsheng (Ross) E. Wang

Rongsheng(Ross) Wang, Ph.D.
Assistant Professor
Department of Chemistry
Temple University

Email: rosswang@temple.edu
Tel: (215) 204-1855
Cell: (314) 341-0544